# Cytoplasmic condensation induced by membrane damage is associated with antibiotic lethality

Felix Wong [1,2], Jonathan M. Stokes[1,2], Bernardo Cervantes[1,2,3], Sider Penkov[4], Jens Friedrichs[5], Lars D. Renner [5✉] & James J. Collins [1,2,6✉]

Bactericidal antibiotics kill bacteria by perturbing various cellular targets and processes. Disruption of the primary antibiotic-binding partner induces a cascade of molecular events, leading to overproduction of reactive metabolic by-products. It remains unclear, however, how these molecular events contribute to bacterial cell death. Here, we take a single-cell physical biology approach to probe antibiotic function. We show that aminoglycosides and fluoroquinolones induce cytoplasmic condensation through membrane damage and subsequent outflow of cytoplasmic contents as part of their lethality. A quantitative model of membrane damage and cytoplasmic leakage indicates that a small number of nanometer-scale membrane defects in a single bacterium can give rise to the cellular-scale phenotype of cytoplasmic condensation. Furthermore, cytoplasmic condensation is associated with the accumulation of reactive metabolic by-products and lipid peroxidation, and pretreatment of cells with the antioxidant glutathione attenuates cytoplasmic condensation and cell death. Our work expands our understanding of the downstream molecular events that are associated with antibiotic lethality, revealing cytoplasmic condensation as a phenotypic feature of antibiotic-induced bacterial cell death.

---

[1] Institute for Medical Engineering & Science and Department of Biological Engineering, Massachusetts Institute of Technology, Cambridge, MA, USA. [2] Infectious Disease and Microbiome Program, Broad Institute of MIT and Harvard, Cambridge, MA, USA. [3] Microbiology Graduate Program, Massachusetts Institute of Technology, Cambridge, MA, USA. [4] Institute for Clinical Chemistry and Laboratory Medicine at the University Clinic and Medical Faculty of TU Dresden, Dresden, Germany. [5] Leibniz Institute of Polymer Research and the Max Bergmann Center of Biomaterials, Dresden, Germany. [6] Wyss Institute for Biologically Inspired Engineering, Harvard University, Boston, MA, USA. ✉email: renner@ipfdd.de; jimjc@mit.edu

Understanding the detailed mechanisms of action of antibiotics remains a central focus in microbiology[1]. While the primary binding partners of antibiotics have been largely classified, we still lack a holistic understanding of how bactericidal antibiotics lead to cell death[2]. Such an understanding would better inform the use and misuse of antibiotics, in addition to approaches for addressing antibiotic resistance.

Previous work has shown that the intracellular accumulation of promiscuously reactive metabolic by-products induces widespread cellular dysfunction and contributes to antibiotic lethality[2–13]. However, the mechanisms underlying how reactive metabolic by-products, and the accompanying phenotypic changes, contribute to cell death have remained unclear. In contrast to conventional approaches that study bacterial responses to antibiotics in bulk culture, we reasoned that analyses of single cells through a physical biology approach[14–16] could allow for a more nuanced understanding of how cell death occurs downstream of the primary drug–target interaction. To this end, we investigated in *Escherichia coli* the single-cell response to three main classes of bactericidal antibiotics: aminoglycosides, which corrupt mRNA proofreading by the 30S ribosomal subunit; fluoroquinolones, which perturb DNA replication; and β-lactams, which disrupt peptidoglycan cell wall biosynthesis through perturbation of penicillin-binding proteins. Here, we show that, in contrast to the lytic mode of action of β-lactams, aminoglycosides and fluoroquinolones induce a phenotype of cytoplasmic condensation through membrane damage. This phenotype is associated with the accumulation of reactive metabolic by-products, lipid peroxidation, and cell death, thus indicating a sequence of cellular events through which reactive metabolic products can lead to antibiotic-induced cell death.

## Results

### Aminoglycosides and fluoroquinolones induce membrane damage and cytoplasmic condensation in *E. coli*.
To probe how aminoglycoside, fluoroquinolone, and β-lactam antibiotics kill metabolically active bacterial cells, we treated log-phase *E. coli* with the aminoglycoside kanamycin, the fluoroquinolone ciprofloxacin, and the β-lactam ampicillin at various concentrations ranging from 1× to 100× the minimum inhibitory concentration (MIC; Supplementary Table 1). Using long-term microfluidic imaging across these conditions, we observed variability in the morphological dynamics of cells, which, aside from filamentation, could be characterized by three modes: cytoplasmic condensation, membrane bulging, and lysis (Fig. 1a and Supplementary Movies 1–3). Interestingly, some cells treated with kanamycin and most cells treated with ciprofloxacin, but no cells treated with ampicillin, exhibited cytoplasmic condensation, a phenotype in which the cytoplasm fragments and grows denser, as manifested by the appearance of discrete, phase-light regions between phase-dark regions that grow darker (Supplementary Figs. 1 and 2, and Supplementary Note 1). Since previous work has explored how β-lactam-mediated perturbation of peptidoglycan biosynthesis leads to membrane bulging and lysis[17], we ventured to explore further in this work the condensation phenotype induced by aminoglycosides and fluoroquinolones.

In cells that have exhibited cytoplasmic condensation, which comprised ~30% of cells treated with kanamycin and ~90% of cells treated with ciprofloxacin, as distinguished visually from long-term timelapses (Fig. 1d), discrete portions of cells typically became phase-light over a timescale of ~1 min. This phenotype characteristically appeared after tens of minutes, or hours, after antibiotic exposure (Fig. 1d), with larger antibiotic concentrations typically resulting in earlier condensation (Supplementary Movies 2 and 3). Subsequent imaging with fluorescent tags

labeling the inner membrane, outer membrane, cytoplasm, and periplasm revealed that the cytoplasm of these cells was plasmolyzed: the inner membrane was retracted from the outer membrane and the periplasmic space corresponding to phase-light regions was increased (Fig. 1b, c), similar to osmotically shocked cells[18]. Supporting the notion that the cytoplasmic contents of these cells are compactified, fluorescence microscopy measurements with a strain exhibiting cytoplasmic fluorescence indicated increased fluorescence intensities in condensed, phase-dark regions (Supplementary Fig. 2). Minutes to hours after the occurrence of condensation, most condensed cells (~90%) eventually lysed (Fig. 1d), as determined by the cytoplasm becoming entirely phase-light. Emphasizing the generality of these phenotypes, we found that cytoplasmic condensation and lysis were shared among different aminoglycosides and fluoroquinolones (Supplementary Fig. 3) and across different antibiotic concentrations (Supplementary Fig. 4), as well as in the Gram-positive bacterium *Bacillus subtilis* during antibiotic treatment, for which we observed that the cytoplasm appeared to retract from the cell wall (Supplementary Fig. 3).

Cytoplasmic condensation and lysis typically require loss of cellular turgor or membrane integrity[17], which are distinct from the primary mechanisms of action of aminoglycosides and fluoroquinolones. Indeed, under standard laboratory conditions, *E. coli* maintains a turgor pressure of ~1 atm which presses its inner membrane against the cell wall[19], raising the question of how condensation of the cytoplasm could physically occur. We hypothesized that the observed condensation and lysis were caused by loss of turgor. To test this hypothesis, we independently: (i) osmotically shocked the cells using flow of hyperosmotic media (Fig. 1e and Supplementary Fig. 5); and (ii) probed the concentration of potassium, an intracellular solute, using a membrane-permeable potassium-sensitive dye, ION Potassium Green (IPG; Supplementary Fig. 6). Quantifying the cellular response to a 500 mM hyperosmotic shock—corresponding to approximately ten times *E. coli*'s estimated turgor pressure[19]—revealed that condensed and lysed cells exhibited smaller length contractions compared to turgid ones (Fig. 1e and Supplementary Fig. 5). As the cell envelopes of cells with larger turgor pressures are more stretched[17], this observation indicates that the turgor of condensed and lysed cells was significantly diminished compared to turgid cells, and that cytoplasmic condensation could occur through loss of cellular turgor (Supplementary Note 1). These observations are consistent with measurements of IPG fluorescence, which we found to be decreased in condensed cells (Supplementary Fig. 6), suggesting that the loss of cellular turgor is associated with the depletion of intracellular solutes including potassium ions.

To better understand how the mechanical properties of cells change during condensation, we next quantified the effective elastic moduli of cells using atomic force microscopy (AFM; Fig. 1f and Supplementary Fig. 7). AFM experiments on condensed, antibiotic-treated cells revealed that, in ciprofloxacin-treated cells, the phase-dark regions of condensed cells exhibit significantly larger elastic moduli than corresponding regions in turgid cells, which, in turn, exhibit similar elastic moduli to phase-light regions (Fig. 1f and Supplementary Fig. 7). As *E. coli*'s turgor pressure has been estimated to be ~1 atm[19], consistent with inferred values based on our AFM measurements (see Methods for details), these observations suggest that decreases in effective stiffness, arising from the loss of turgor pressure during cytoplasmic condensation, may be offset by the intracellular compactification of macromolecules suggested by phase-contrast and fluorescence microscopy measurements (Supplementary Fig. 2). Intriguingly, we observed similar, but less pronounced, variations in cell stiffness in kanamycin-treated cells, whose cell-

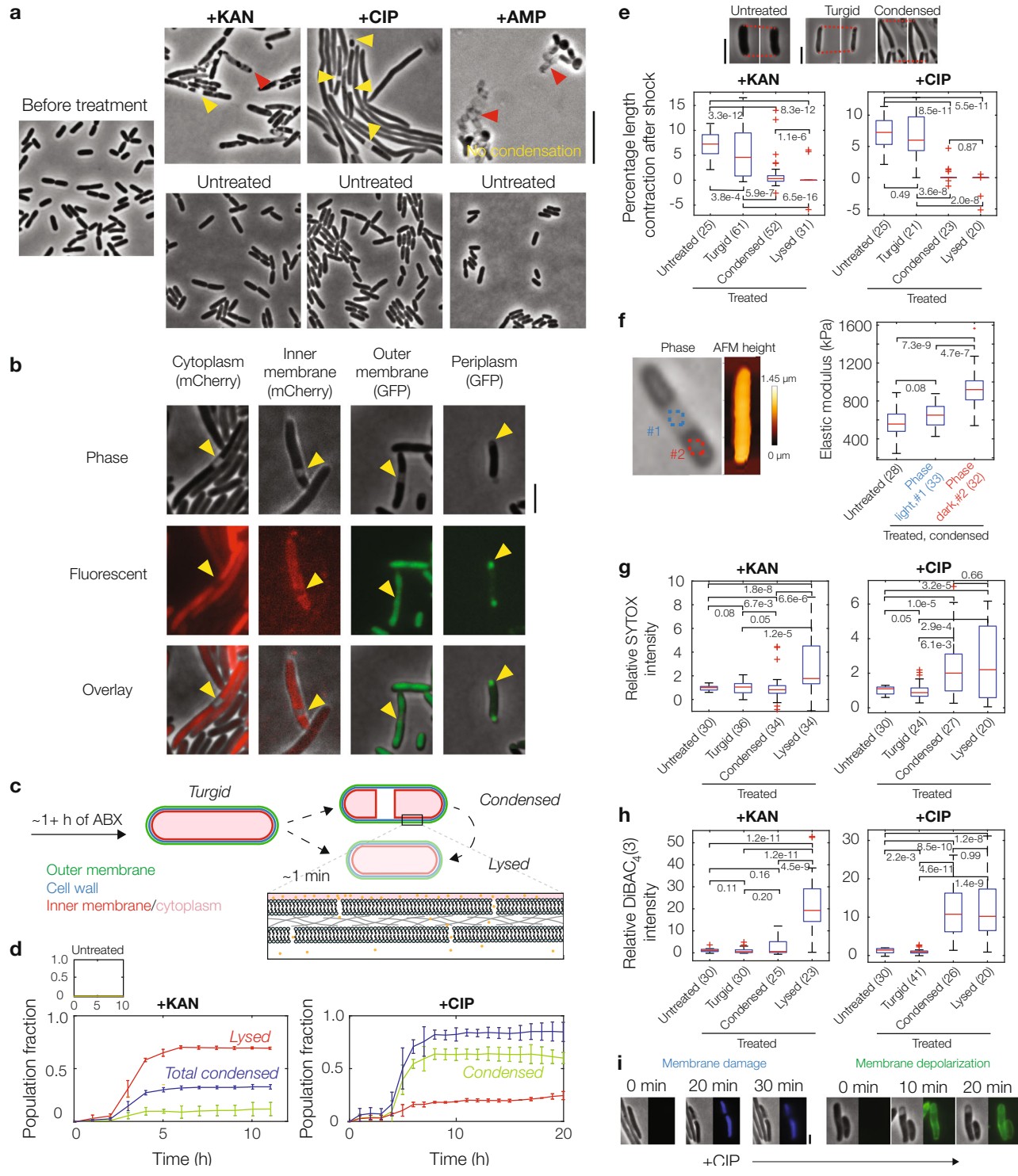

averaged elastic moduli are not significantly changed relative to those of untreated cells (Supplementary Fig. 7). It is possible that this difference between condensed ciprofloxacin- and kanamycin-treated cells could arise from additional membrane damage through ribosome-independent ionic interactions of aminoglycosides with the outer membrane, which contributes to destabilizing the cellular envelope[20]. Another possibility suggested by our fluorescence intensity measurements (Supplementary Fig. 2) is that, while cellular turgor is decreased in condensed cells treated with either antibiotic, less intracellular compactification of macromolecules occurs in kanamycin-treated cells. Thus, these

AFM experiments suggest that cytoplasmic condensation affects the effective elastic moduli of antibiotic-treated cells to varying degrees.

We next investigated whether the observed loss of turgor arose specifically from membrane damage, resulting in the outflow of cytoplasmic contents. Addition of SYTOX Blue, a DNA intercalating dye that only penetrates cells with compromised membranes, to drug-treated cells indicated loss of membrane integrity in condensed and lysed cells, with kanamycin-treated cells exhibiting significant increases in fluorescence at lysis and ciprofloxacin-treated cells exhibiting significant increases in

**Fig. 1 Aminoglycoside and fluoroquinolone antibiotics induce cytoplasmic condensation in *E. coli*. a** Cytoplasmic condensation and/or lysis (yellow and red markers, respectively) induced by kanamycin (KAN), ciprofloxacin (CIP), and ampicillin (AMP) after 1–6 h of treatment (KAN: 3 h, CIP: 6 h, AMP: 1 h) in a population of wild-type, log-phase *Escherichia coli* cells, imaged in phase contrast. Times were chosen to reflect timescales of phenotypic change. Here and below, all antibiotic concentrations used were 10× MIC, while control cells were untreated; working MICs were 5 µg/mL for kanamycin, 0.1 µg/mL for ciprofloxacin, and 10 µg/mL for ampicillin. Ampicillin-treated cells exhibited membrane bulging and lysis, in contrast to cytoplasmic condensation and lysis observed in kanamycin- and ciprofloxacin-treated cells (Supplementary Fig. 1). Ciprofloxacin-treated cells also displayed filamentation. Cytoplasmic condensation and lysis were general across cells in a population, antibiotic concentrations ranging at least from 1× to 100× MIC, different aminoglycoside and fluoroquinolone antibiotics, and in *Bacillus subtilis* (Supplementary Figs. 3 and 4). Results are representative of three biological replicates in each condition. Scale bar, 10 µm. **b** *E. coli* cells with fluorescent genetic outer membrane (green fluorescent protein, GFP), inner membrane (mCherry), periplasmic (GFP), and cytoplasmic (mCherry) markers under ciprofloxacin treatment, imaged in phase contrast and epifluorescence. For similar cells treated with kanamycin, see Supplementary Fig. 3. Results are representative of three biological replicates for each marker, and yellow markers highlight condensed cells. Scale bars here and below, 3 µm. **c** Schematic of cytoplasmic condensation and lysis in *E. coli* treated with kanamycin and ciprofloxacin. **d** Fractions of all cells that are condensed or lysed. Error bars indicate one standard deviation, and data are presented as mean ± SEM. Data are derived from two different fields of view with at least 100 cells each (kanamycin, 133 and 149 cells; ciprofloxacin, 255 and 140 cells) and representative of two biological replicates. Note that cells may condense and lyse within the same hour; such cells are reflected only in lysed cell counts. The total condensed fraction counts all cells that are, or have previously been, condensed. **e** Hyperosmotic shock measurements in control and antibiotic-treated *E. coli*, of which three characteristic phase-contrast microscopy images are shown (top); here, ciprofloxacin-treated cells are shown. The percentage length contraction, an indicator of cellular turgor, is the percentage change in cell length immediately after hyperosmotic shock. Red dashed lines represent visual guides. The number of cells in each group are indicated in parentheses. Here and below, a box plot indicating the median (center), 25th and 75th percentile (bounds of box), and extreme data points not considered outliers (bounds of whiskers) is shown, and red crosses indicate outliers including the minimum and maximum values. *p*-values for two-sample Kolmogorov-Smirnov tests are shown next to corresponding brackets. **f** Atomic force microscopy (AFM) of cells. (Left) Phase-contrast and AFM topography image of a condensed, ciprofloxacin-treated cell. (Right) Inferred elastic moduli corresponding to the cellular regions shown to the left, as indicated by different colors, as well as along the cellular body of an untreated control cell. The number of cells in each group are indicated in parentheses. Box plot features and statistical tests refer to those of panel (**e**). For measurements for kanamycin-treated cells, see Supplementary Fig. 7. **g, h** Fluorescence intensities of control and antibiotic-treated *E. coli* in the presence of SYTOX Blue (**g**) and DiBAC$_4$(3) (**h**). The number of cells in each group are indicated in parentheses. Box plot features and statistical tests refer to those of panel (**e**). **i** Representative timelapses of ciprofloxacin-treated *E. coli* stained by SYTOX Blue and DiBAC$_4$(3), imaged in phase contrast and epifluorescence. Times show duration after the first image, and results are representative of three biological replicates.

fluorescence during condensation (Fig. 1g, i and Supplementary Fig. 8); we speculate that this difference may arise from varying levels of membrane damage, as well as earlier lysis after condensation in kanamycin-treated cells (Supplementary Fig. 9), which might allow for limited intercalation of dyes in these cells before lysis. Application of the membrane potential-sensitive dye DiBAC$_4$(3) similarly indicated membrane depolarization concomitant with membrane damage (Fig. 1h, i and Supplementary Fig. 8). Consistently, as a control, fluorescence of SYTOX Blue and DiBAC$_4$(3) did not occur for plasmolyzed cells that were subjected to a 500 mM hyperosmotic shock, for which no cells lysed (Supplementary Fig. 8). Taken together, these results reveal that aminoglycosides and fluoroquinolones can induce cytoplasmic condensation and lysis through membrane damage, and, as suggested by SYTOX Blue, this membrane damage coincides with loss of cell viability. As detailed below, further time-kill assays confirm that the observed membrane damage accompanies loss of cell viability.

**A biophysical model of cell envelope mechanics and solute outflow predicts the dynamics of cytoplasmic condensation.** Having determined that cytoplasmic condensation occurs through antibiotic-induced membrane damage, we sought to better understand the extent of membrane damage and the physiology of condensed cells using biophysical modeling. To this end, building on previous work[17,21,22], we developed a model of cell envelope mechanics that predicts: (i) smaller turgor and cytoplasmic condensation to arise from the elastic relaxation to an equilibrium state, one which is governed by the outward flow of cytoplasmic solutes through nanometer-sized membrane defects; and (ii) the number of such defects consistent with the empirically observed ~1 min timescale and magnitude of cytoplasmic condensation (Fig. 2a–c, Supplementary Note 1, Supplementary Table 2, and Supplementary Fig. 10).

The model assumes, for simplicity, that a constant number of nanometer-scale defects are introduced into both the inner and outer membranes of a bacterium. These defects need not overlap across the cell envelope layers, and similar defects need not be introduced into the peptidoglycan cell wall because the cell wall is porous with characteristic pore areas of ~10 nm$^2$ [23]. While similar membrane defects may be generated, and quickly alleviated, during processes including electroporation, the SYTOX Blue (Fig. 1g) and osmotic shock experiments (Fig. 1e) described above suggest that the membrane defects we consider are sustained across time, so that appreciable outflow of intracellular solutes and intercalation of fluorescent dyes occur. Based on a linear-elastic description of the cellular envelope and a simple model of fluid transport through Poiseuille flow, the model predicts that the hydrodynamic flow of solutes from the cytoplasm to the external milieu is laminar and results in a gradual loss of turgor (Fig. 2d, e). This loss of turgor decreases the mechanical strains in the cell envelope. In turn, solute outflow is attenuated because both the cellular turgor and the sizes of typical membrane nanodefects are decreased. Cytoplasmic condensation occurs as cell volume is lost (Fig. 2f and Supplementary Fig. 9), and the model predicts that condensation corresponds to a steady state of this dynamical system. Surprisingly, the model predicts that only 10 nanometer-scale defects, corresponding to loss of approximately 1 in 10$^6$ of all membrane phospholipids, can quantitatively explain both the timescale and magnitude of cytoplasmic condensation (Fig. 2f). Thus, building on our empirical measurements, the model reveals that subtle changes to membrane integrity can provoke the cellular-scale phenotype of cytoplasmic condensation.

**Cytoplasmic condensation is associated with cell death.** To better understand the significance of membrane damage and accompanying cytoplasmic condensation in a more conventional experimental system, we performed single-cell analyses and

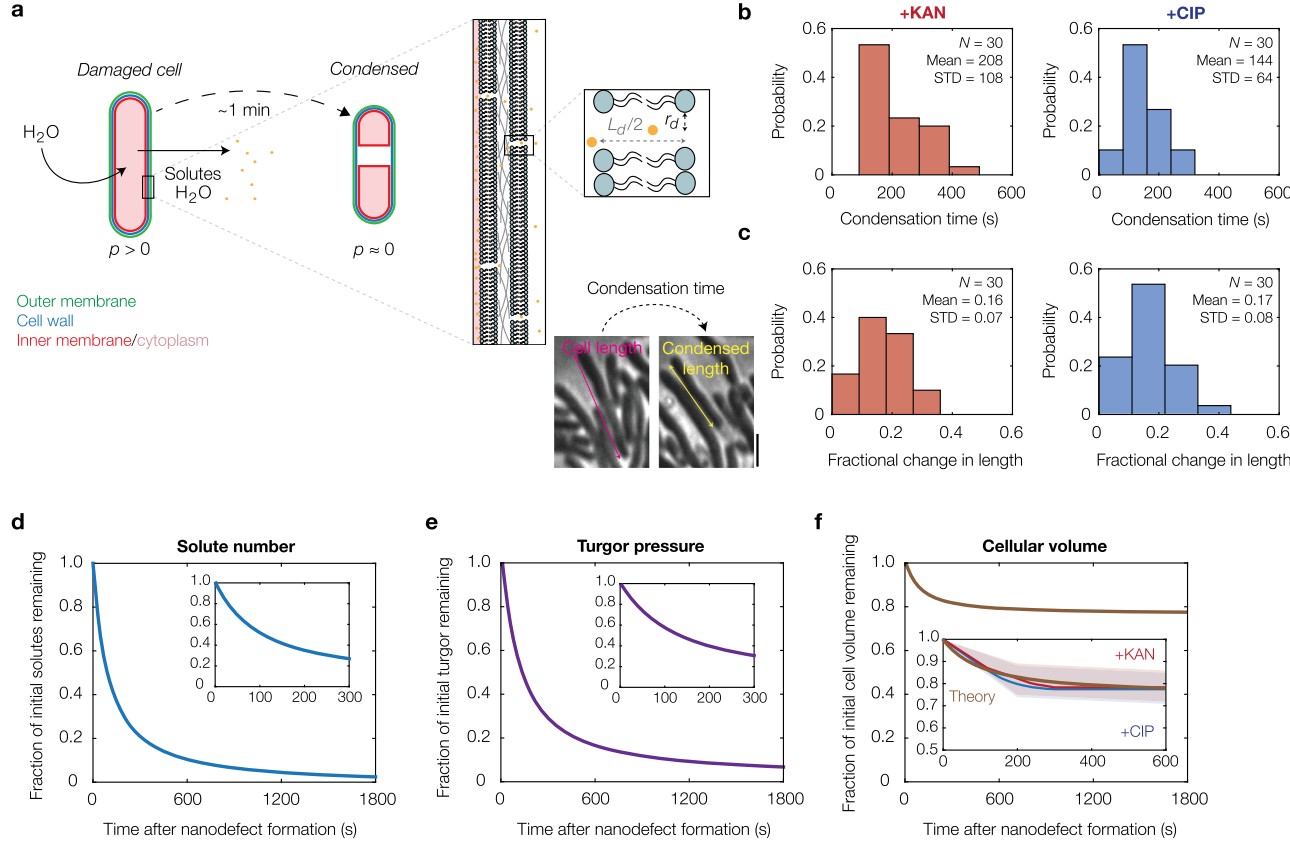

**Fig. 2 Cytoplasmic condensation emerges from subtle, nanometer-scale membrane defects. a** Model of condensation due to membrane nano-defect formation and subsequent solute leakage. Membrane nano-defects are modeled as circular channels that extrude through both the inner and outer membranes, with a typical radius of $r_d$ and a typical length of $L_d$. Cytoplasmic condensation occurs as the cellular turgor ($p$) decreases. For details of the model, see Supplementary Note 1. **b** Empirically observed distributions of condensation times in kanamycin-treated and ciprofloxacin-treated cells at 10× MIC. The cell numbers ($N$), population means, and standard deviations (STD) are indicated. **c** Same as panel (**b**) but for distributions of cell length changes, as illustrated in phase contrast for a typical ciprofloxacin-treated cell (10× MIC) in the inset (left; scale bar, 3 μm). **d–f** Model predictions for solute number (**d**), turgor pressure (**e**), and cell volume (**f**) against time after membrane nano-defect formation, for the characteristic parameter values summarized in Supplementary Table 2. (Insets) Plots for short times. The empirically observed decreases in cellular volume as functions of time for the kanamycin-treated and ciprofloxacin-treated (10× MIC) cells, represented in part by panels (**b**) and (**c**), are indicated by red and blue curves, respectively. Shaded regions indicate one standard deviation.

concurrent bulk time-kill assays. These revealed that condensation and lysis events occur in time concomitant with >1 log of killing in culture (Fig. 3a), as determined after plating and overnight incubation by colony-forming unit (CFU) quantification. We note here that decreases in CFUs were observed as soon as ~1 h after antibiotic treatment (Fig. 3a); however, this observation does not imply that cell death occurs before cytoplasmic condensation, since cell death may occur after plating in corresponding time-kill assays. Consistent with this hypothesis, recent work from the Zhao and Drlica labs has indicated that antibiotic-induced cell death occurs after plating on drug-free agar due, in part, to the post-stress accumulation of reactive oxygen species (ROS)[11], so that cell death may occur significantly after plating.

To further probe the notion that cells die on timescales longer than ~1 h of antibiotic treatment, we performed bacterial growth measurements at the single-cell level. These measurements showed that, in cells treated with ciprofloxacin, condensation represented an irreversible step at which cells failed to elongate and filament, with cessation of filamentation occurring at the time of condensation (Fig. 3b, c). Interestingly, kanamycin-treated cells exhibited increased growth variability before condensation and oscillatory cell length dynamics (Fig. 3d, e), consistent with repeated membrane swell–burst cycles[20] and

suggestive of more heterogeneous membrane damage compared to ciprofloxacin-treated cells. This observation is also consistent with earlier work showing that aminoglycosides induce outer membrane damage by ribosome-independent ionic interactions, which, along with subsequent membrane fusion, lead to repeated membrane swell–burst cycles[20]. For both antibiotics, subsequent washout with drug-free growth media revealed that condensed and lysed cells do not recover from antibiotic treatment, while filamentous or dividing cells recover from antibiotic treatment and appear to proliferate (Fig. 3f, g and Supplementary Movies 4 and 5). These results indicate that the cessation of filamentation, seen in condensed cells, is associated with killing (Fig. 3b, c). Furthermore, these observations collectively suggest that cytoplasmic condensation is an irreversible phenotype distinguishing antibiotic susceptibility from tolerance, although we note here that, because antibiotic treatment leads to >1 log of killing but less than 90% of cells observed may be condensed, not all susceptible cells are condensed (Figs. 1d and 3a).

Unsurprisingly, experiments in which drug-tolerant stationary-phase cells were treated with kanamycin and ciprofloxacin revealed less killing in culture, and less cytoplasmic condensation and lysis in single cells (Supplementary Fig. 11). This is expected because recent work has shown that the lethality of different

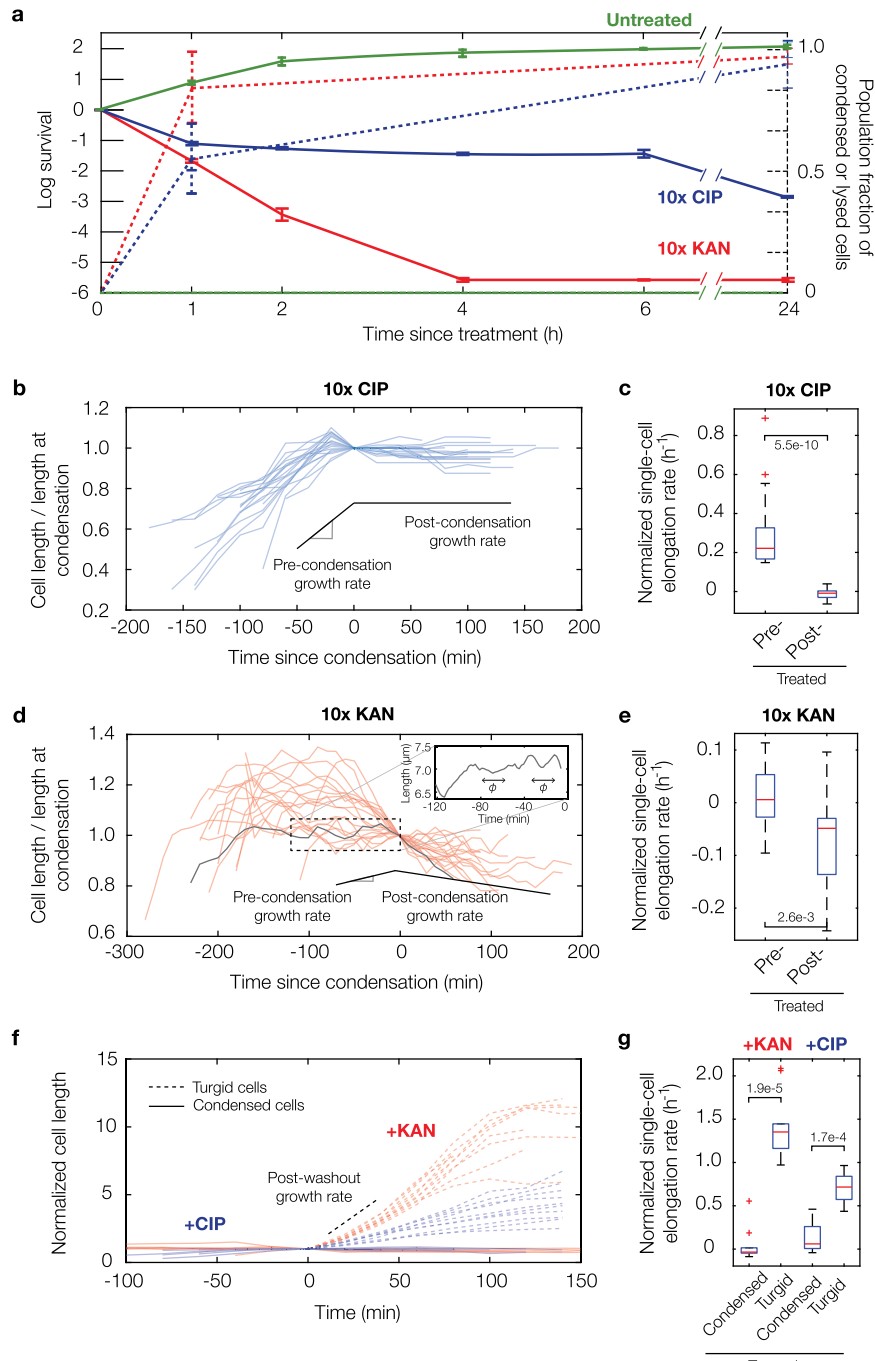

**Fig. 3 Cytoplasmic condensation is an irreversible cell death phenotype. a** Survival of log-phase bulk cultures of *E. coli* treated with kanamycin and ciprofloxacin at 10× MIC and corresponding fractions of condensed and lysed cells after treatment for 24 h. Each timepoint represents two biological replicates and, for population fractions, two different fields of view with at least 100 cells each (untreated, 109 and 100 cells; kanamycin, 113 and 102 cells; ciprofloxacin, 175 and 120 cells). For population fractions, a population of cells was treated with antibiotics for the indicated treatment times inside a microfluidic chamber, cells were washed with drug-free medium, and the number of condensed and lysed cells after 20 h were counted. Error bars indicate one standard deviation, and data are presented as mean ± SEM. **b** *E. coli* cell length (*L*) traces of 20 cells treated continuously with ciprofloxacin at 10× MIC inside a microfluidic chamber. **c** Average normalized single-cell elongation rates (*dL/(Ldt)*) of cells corresponding to panel (**b**), with pre-condensation (pre-) and post-condensation (post-) values shown. Here and below, box plots indicating the median (center), 25th and 75th percentile (bounds of box), and extreme data points not considered outliers (bounds of whiskers) are shown, and red crosses indicate outliers including the minimum and maximum values. *p*-values for two-sample Kolmogorov-Smirnov tests are shown next to corresponding brackets. **d–e** Same as (**b**, **c**), but for treatment with kanamycin at 10× MIC, for which cells exhibited cell length oscillations approximately described by a typical period of *ϕ* = 20 min (inset). Box plot features refer to those of panel (**c**). **f**, **g** Same as (**b**, **c**), but for transient antibiotic treatment and subsequent washout, showing that recovering, turgid cells survive and continue to elongate. Here, time is defined as time since condensation for condensed cells and time since start of measurement for recovering cells. Cell length represents cumulative cell length of a cell and, if the cell divides, all its progeny. The length is normalized to its value at condensation for condensed cells and to its starting length for recovering cells. All cells were treated for 30 min with kanamycin or ciprofloxacin at 10× MIC inside a microfluidic chamber. Box plot features refer to those of panel **c**.

antibiotics, including aminoglycosides and fluoroquinolones, is associated with cellular metabolism[24,25]. As cellular metabolism is decreased in stationary-phase cells (Supplementary Fig. 12), the decreased frequencies of condensation and lysis—in addition to decreased killing—observed in these cells are consistent with the finding that these cells are less susceptible to antibiotics.

**Cytoplasmic condensation is associated with ROS accumulation, lipid peroxidation, and changes in lipid composition.** Although membrane damage and cytoplasmic condensation occur rapidly after antibiotic exposure, inhibition of DNA replication and protein translation, by themselves, are not known to lead to membrane damage, prompting our inquiry into the underlying mechanism of these phenotypes. While aminoglycosides have been shown to perturb membranes through non-specific ionic bonds and misfolded proteins[26–28], another possibility that may explain our observations with both aminoglycosides and fluoroquinolones is that promiscuously reactive metabolic by-products damage membrane phospholipids through processes involving, for example, free-radical reactions[2,3,13,29]. Previous work has shown that antibiotic-treated cells experience increased flux through metabolic pathways that lead to reactive metabolic by-products irrespective of the primary antibiotic-binding target, and these reactive metabolic by-products have been evidenced to contribute to antibiotic lethality[2–13,29]. Indeed, consistent with this notion, we observed that condensed and/or lysed antibiotic-treated cells exhibited significantly elevated levels of reactive oxygen species (ROS) and reactive nitrogen species (RNS), as measured by the dyes carboxy-$H_2$DCFDA and DAF-FM (Fig. 4a, b).

As such, we wondered whether cytoplasmic condensation specifically correlated with lipid peroxidation, a process in which free radicals and other reactive molecules react with fatty acids and may ultimately rupture the membrane[30]. To test this hypothesis, we assayed lipid peroxidation both in single cells and bulk culture using the lipophilic dye C11-BODIPY. The fluorescence of this dye shifts from red to green when oxidized by oxy-radicals and peroxynitrite[31], and measurements of green fluorescence intensity over time offer a quantitative measure of lipid peroxidation dynamics. Consistent with the hypothesis that cytoplasmic condensation correlates with lipid peroxidation, we observed that condensed and lysed cells exhibited larger intensities than turgid cells (Fig. 4c). Moreover, cells exhibited sharp increases in fluorescence coincident in time with condensation and lysis (Fig. 4d). Flow cytometry measurements using bulk cultures also confirmed an increase in fluorescence, implying increases in ROS and RNS, consistent with single-cell measurements (Fig. 4e–g).

As the foregoing results indicate changes in membrane composition, we next characterized the lipid contents of antibiotic-treated cells (Fig. 5). We found that bulk cultures treated with kanamycin and ciprofloxacin, in addition to bulk cultures treated with ampicillin and 10 mM $H_2O_2$ considered here for comparison, exhibited detectable increases in free fatty acids (FFA) relative to untreated cells (Fig. 5a). In particular, averaged measurements of peak area intensities suggest that FFA levels are increased by as much as ~1-fold in kanamycin and ciprofloxacin-treated cells and by as much as ~7-fold in ampicillin-treated cells, with the size of the increase varying with treatment time (Fig. 5b). While the latter may be expected based on the lytic mode of action of ampicillin, the observation of increased FFA levels in kanamycin- and ciprofloxacin-treated cells contrasts with the primary modes of action of aminoglycosides and fluoroquinolones and is consistent with the membrane damage characterized above using SYTOX Blue (Fig. 1g). We also observed salient shifts

in cardiolipin (CL)/phosphatidylethanolamine (PE) ratios (Fig. 5a, c), which have previously been shown to be associated with changes in cell envelope ultrastructure similar to that found for cytoplasmic condensation here[32]. Indeed, we found that CL/PE ratios could be ~2 to ~3-fold higher in kanamycin- and ciprofloxacin-treated cells, suggesting that antibiotic treatment and subsequent cytoplasmic condensation may be generally associated with changes in the CL/PE ratio and lipid composition. In sum, these results are consistent with our observations of membrane damage, and together our experiments demonstrate that antibiotic-induced cytoplasmic condensation is associated with the accumulation of promiscuously reactive metabolic by-products (Fig. 4), lipid peroxidation (Fig. 4), and changes in lipid composition (Fig. 5).

Finally, as previous work has shown that excess lipopolysaccharide (LPS) can lead to cell death in a *mla** strain of *E. coli*[33], we asked as well whether changes in LPS levels are associated with the cell death phenotype characterized here. Measuring LPS concentrations in antibiotic-treated bulk cultures with a limulus amebocyte lysate (LAL) assay, we found, intriguingly, that kanamycin- and ciprofloxacin-treated cultures exhibited significantly lower LPS concentrations than did untreated cultures (Supplementary Fig. 13). This observation suggests that increased LPS levels may not be associated with cell death in cells treated with kanamycin and ciprofloxacin.

**Cytoplasmic condensation and antibiotic-induced cell death are mitigated by glutathione, a scavenger of reactive metabolic by-products.** We next sought to determine whether the reactive metabolic by-products that are associated with cytoplasmic condensation also contribute to antibiotic lethality, as measured at the single-cell level by the frequency of condensation and lysis as well as in bulk culture by time-kill assays. To this end, we pre-treated growing cultures of *E. coli* with glutathione, an antioxidant and scavenger of reactive metabolic by-products that is also known to attenuate antibiotic killing of bulk cultures[13,24]. We observed significantly decreased killing by both kanamycin and ciprofloxacin in glutathione-pretreated cultures (Fig. 6a, b). Specifically, exogenous supplementation of glutathione (10 mM) decreased kanamycin killing by as much as ~7 logs and ciprofloxacin killing by as much as ~4 logs for cell cultures treated at the previously determined MICs. As glutathione also increases the MICs of kanamycin and ciprofloxacin (Supplementary Table 1), we performed further time-kill assays relative to these increased MICs (Fig. 6a, b). These assays revealed increased survival relative to cases of no glutathione with kanamycin treatment in the range of 1× MIC and ciprofloxacin treatment at various treatment times in the range of 0.1× to 10× MIC (Fig. 6a, b), suggesting that the glutathione protection observed in bulk cultures may arise from both decreasing growth inhibition and decreasing killing of individual cells. At the single-cell level, antibiotic treatment of glutathione-pretreated cells was accompanied by cell proliferation in kanamycin-treated cells and notable suppression of condensation and lysis in ciprofloxacin-treated cells (Fig. 6c, h and Supplementary Movies 6 and 7), suggesting a reactive metabolic by-product-based origin for the membrane damage and decreased survival observed in the absence of glutathione. Furthermore, while antibiotic lethality correlates with cellular metabolism[24,25,29], we note here that the observed protection from antibiotic killing by glutathione did not arise from differences in intracellular ATP levels, as indicated by bulk-culture measurements of ATP abundance (Supplementary Fig. 12). Indeed, since glutathione is involved not only in ROS and RNS scavenging but also cellular detoxification of additional reactive metabolic by-products such as methylglyoxal[34], these results

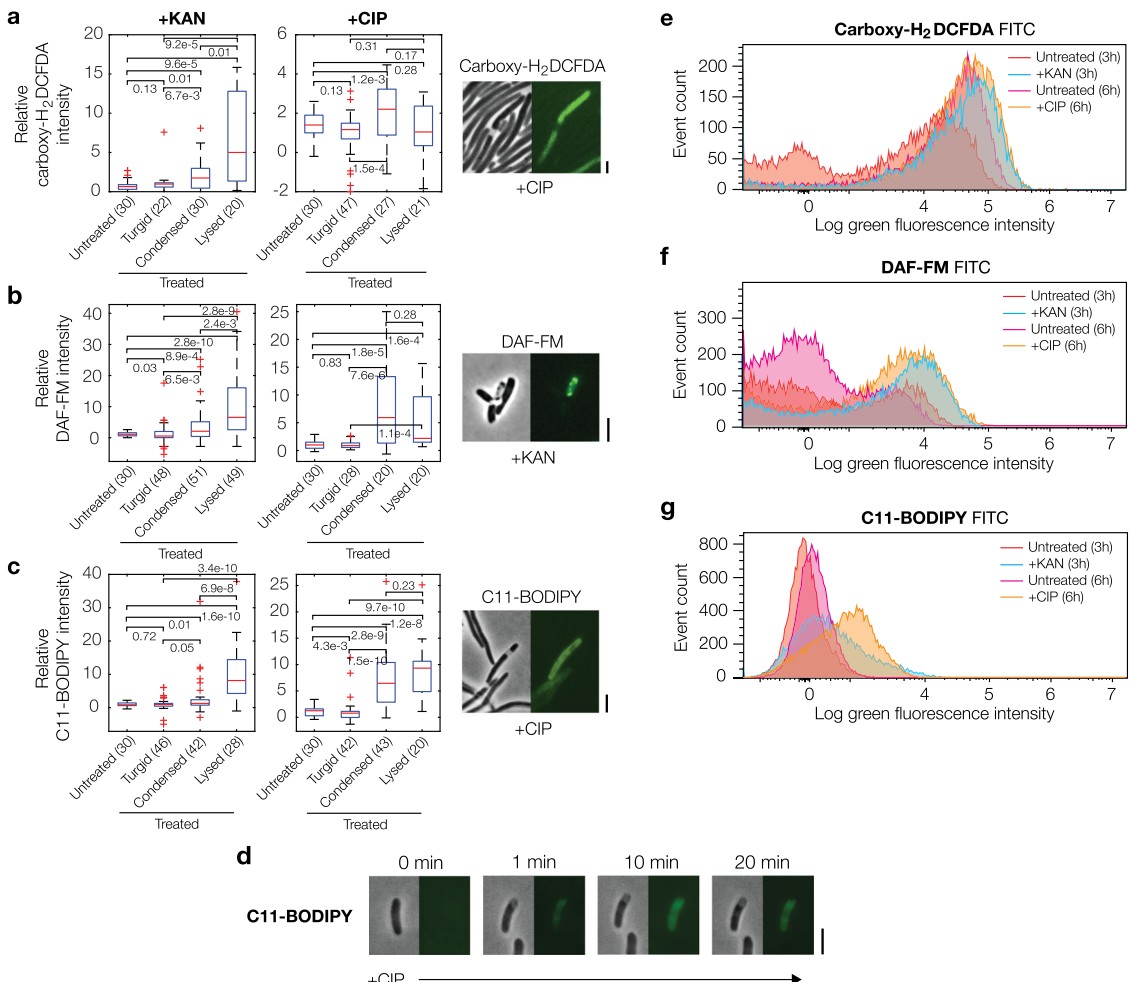

**Fig. 4 Cytoplasmic condensation is associated with the accumulation of reactive metabolic by-products. a** Fluorescence intensities of antibiotic-treated *E. coli* in the presence of the ROS-sensitive dye carboxy-$H_2$DCFDA. A representative view of ciprofloxacin-treated cells imaged in phase contrast and epifluorescence is shown. Here and below, the kanamycin and ciprofloxacin concentrations used were 10× MIC, error bars represent one standard deviation, and scale bars indicate 3 μm. The number of cells in each group are indicated in parentheses. Box plots indicating the median (center), 25th and 75th percentile (bounds of box), and extreme data points not considered outliers (bounds of whiskers) are shown, and red crosses indicate outliers including the minimum and maximum values. *p*-values for two-sample Kolmogorov-Smirnov tests are shown next to corresponding brackets. **b**, **c** Same as (**a**), but for DAF-FM, a dye sensitive to nitric oxide (**b**), and C11-BODIPY, a lipid peroxidation-sensitive dye (**c**). **d** A representative timelapse of a ciprofloxacin-treated *E. coli* cell in the presence of C11-BODIPY. Times show duration after the first image, and results are representative of three biological replicates. **e–g** Histograms of fluorescence intensities of populations of control and antibiotic-treated *E. coli*, as assayed by flow cytometry using the FITC fluorochrome. Dyes and antibiotic concentrations were the same as in (**a–c**), and cells were assayed after 3 h (untreated and kanamycin-treated cells) and 6 h (untreated and ciprofloxacin-treated cells) of treatment. Cells were analyzed after longer times for ciprofloxacin treatment because a majority of condensation events occurred later in time (Fig. 1d). Data are representative of four biological replicates and 20,000 scattering events for each distribution.

suggest that reactive metabolic by-products—which may include, but are not limited to, ROS and RNS—contribute to antibiotic-induced cell death, and that glutathione-mediated detoxification of these molecules contributes to protection from antibiotic lethality.

Next, we performed time-kill experiments on Δ*gor* and Δ*gshA* strains of *E. coli*, which, respectively, lack genes encoding for glutathione reductase (Gor) and a glutamate-cysteine ligase (GshA) contributing to glutathione synthesis[35,36]. We found that antibiotic lethality remained largely unchanged from wild-type controls, suggesting that the modifications to intracellular glutathione pools arising from these genetic deletions have limited effects on aminoglycoside and fluoroquinolone killing (Supplementary Fig. 14). As endogenous pools of glutathione have been estimated to be ~10 mM in *E. coli*[37], it is possible that these genetic perturbations induce smaller variations in glutathione concentration than are needed to protect against

antibiotic lethality resulting from the production of reactive metabolic by-products.

As a positive control for a scavenging reaction involving glutathione, we repeated our time-kill experiments with treatment by exogenous peroxynitrite[38] with and without pretreatment by glutathione (10 mM). To explore the effects of different antioxidants, we also considered the antioxidants dithiothreitol and mercaptoethanol (10 mM). Intriguingly, treating exponentially growing bulk cultures with peroxynitrite resulted in several logs of killing, which was accompanied at the single-cell level by cytoplasmic condensation and lysis (Fig. 6d–f). Yet, while pretreatment with any of the three antioxidants considered rescued cells from peroxynitrite killing and appeared to suppress peroxynitrite-induced cytoplasmic condensation and lysis (Fig. 6e, f, h), we found that cells pretreated with dithiothreitol and mercaptoethanol were not substantially rescued from kanamycin and ciprofloxacin-induced cell death and cytoplasmic

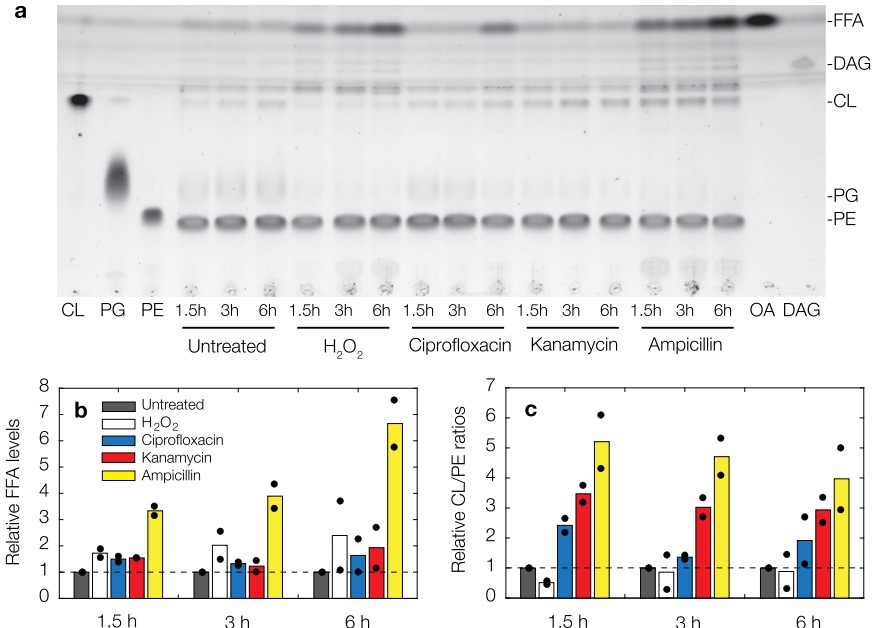

**Fig. 5 Antibiotic treatment is associated with changes in lipid composition. a** Representative thin-layer chromatography image for lysates of control and antibiotic-treated cell cultures. Cells were harvested after various treatment times as shown, and antibiotic concentrations used were 10× MIC. For comparison, results for cells treated with 10 mM hydrogen peroxide and 10× MIC ampicillin are shown. Results are representative of two biological replicates. Abbreviations: CL, cardiolipin; PG, phosphatidylglycerol; PE, phosphatidylethanolamine; FFA, free fatty acids; DAG, diacylglycerol; OA, oleic acid. **b**, **c** Relative FFA levels (**b**) and CL/PE ratios (**c**) across treatment conditions, relative to values for untreated cells at each treatment time. Data are from two biological replicates (black points), and bars indicate averages.

condensation (Fig. 6g, h). Consistent with these observations, kanamycin and ciprofloxacin MICs were unchanged in the presence of dithiothreitol and mercaptoethanol (Supplementary Table 1). These differences, which could arise from varied scavenging potentials toward different reactive metabolic by-products, suggests that aminoglycoside and fluoroquinolone-induced cytoplasmic condensation, lysis, and cell death may not be explained by the accumulation of peroxynitrite alone.

Finally, in order to test whether certain compounds might attenuate lipid peroxidation and cell death in antibiotic-treated cells, we performed microscopy and time-kill experiments involving the exogenous supplementation of α-tocopherol, a lipophilic antioxidant which has been shown to mitigate lipid peroxidation in eukaryotes[39]. We found that pretreating with α-tocopherol (50 mM) did not significantly alleviate antibiotic lethality (Fig. 6g) or alter kanamycin and ciprofloxacin MICs (Supplementary Table 1). Consistent with our characterization of cytoplasmic condensation as a cell death phenotype, cytoplasmic condensation and fluorescence of C11-BODIPY also occurred similarly to cells without α-tocopherol pretreatment (Fig. 6g, h and Supplementary Fig. 15). We speculate that α-tocopherol may be less potent in *E. coli* than in eukaryotes due to potentially limited scavenging potential and uptake. Intriguingly, a recent study developed a lipophilic antioxidant different from α-tocopherol and showed that it prevented lipid peroxidation and fluoroquinolone-induced cell death in *E. coli*[40]; together with our results, this observation suggests a varied landscape of chemical compounds that could attenuate lipid peroxidation and, in doing so, mitigate antibiotic lethality.

**Cytoplasmic condensation and antibiotic-induced cell death are robust to changes in oxygen availability and external osmolarity**. As the foregoing results suggest that cytoplasmic condensation and antibiotic-induced cell death are associated with membrane damage induced by reactive metabolic by-products, we asked how cytoplasmic condensation and antibiotic

lethality could be affected by perturbations including (1) the absence of environmental oxygen; and (2) osmotic shifts that might physically restore cellular turgor and reverse the condensed phenotype. To address this question, we performed microscopy and time-kill experiments.

We first considered the effects of oxygen availability on our findings. We performed microscopy and time-kill experiments in an anaerobic chamber similarly to previous work from our group[13]. Intriguingly, when cells were cultured, treated, and imaged under strictly anaerobic conditions, we found that treatment by kanamycin and ciprofloxacin still induced cyto-plasmic condensation and fluorescence of C11-BODIPY (Fig. 7a). Surprisingly, however, we found that glutathione also conferred protection in anaerobic time-kill assays (Fig. 7b), suggesting that glutathione may be involved in cellular detoxification in the absence of environmental oxygen. Survival of glutathione-pretreated bulk cultures under kanamycin treatment was increased similarly to aerobic conditions, by ~3 logs at 10× the baseline aerobic MIC, while survival under ciprofloxacin treat-ment was increased by ~1 to 2 logs at 1× and 10× MIC (Fig. 7b). Together, these findings indicate that reactive metabolic by-products in addition to ROS, including RNS[13], may contribute to cytoplasmic condensation and antibiotic-induced cell death under anaerobic conditions.

We next performed microfluidic and time-kill experiments in which antibiotic-treated cells were hypoosmotically shocked, in order to produce increases in cellular turgor. In microfluidic experiments, cells were grown in LB with 250 mM sorbitol before and during ~3 h of treatment with kanamycin or ciprofloxacin (at 10× MIC). At the onset of cytoplasmic condensation, cells were hypoosmotically shocked with a flow of sorbitol-free, drug-containing LB. In untreated controls, this hypoosmotic shock results in the transient enlargement of cells due to increased turgor, with typical cytoplasmic volume increases of less than ~5%[41], as confirmed here (Supplementary Movie 8). In most antibiotic-treated cells, we found that this

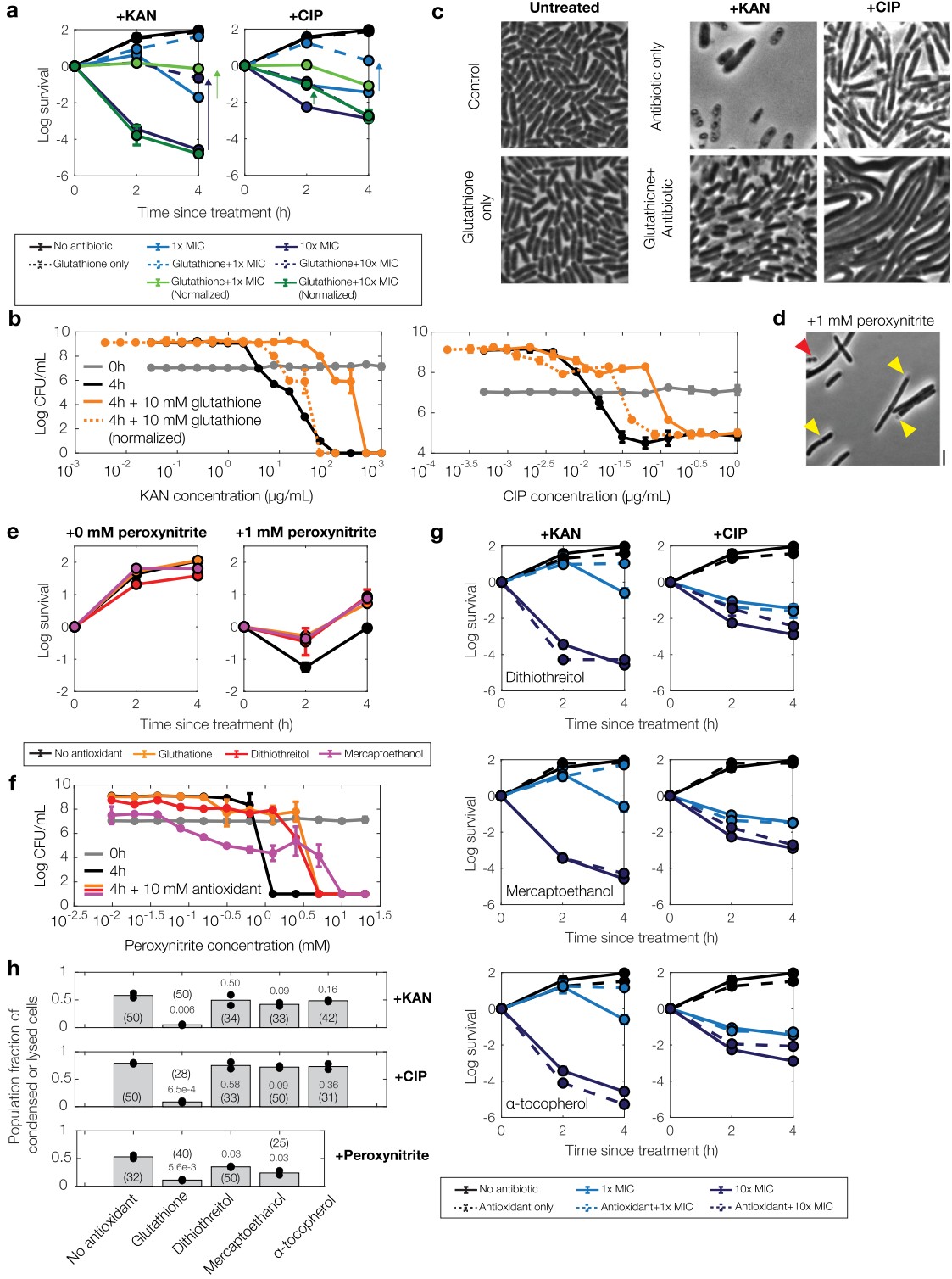

hypoosmotic shock did not change the condensed phenotype. Nevertheless, we found that a small fraction (~10%) of condensed, ciprofloxacin-treated cells briefly became turgid, as indicated in phase-contrast by phase-light regions becoming phase-dark and increases in cell size, so that hypoosmotic shock appeared to transiently reverse cytoplasmic condensation in these cells (Fig. 7c and Supplementary Movie 9). Yet, after several minutes—a timescale consistent with the timescale of solute outflow predicted by our model—these cells reverted to the condensed phenotype and failed to elongate upon

continued observation. In contrast to ciprofloxacin-treated cells, we observed typical kanamycin-treated cells to lyse immediately after hypoosmotic shock (Supplementary Movie 10), as would be consistent with a scenario in which kanamycin-induced membrane damage is more widespread—potentially due to ribosome-independent ionic interactions with the outer membrane[20,26]—and results in a lower resistance of the membranes to stretching. Importantly, for both antibiotics, application of hypoosmotic shock failed to rescue cells by resulting in continued cell growth.

**Fig. 6 Cytoplasmic condensation and antibiotic killing vary with the supplementation of glutathione, an antioxidant, but not dithiothreitol, mercaptoethanol, or α-tocopherol. a, b** Survival curves of *E. coli* under kanamycin and ciprofloxacin treatment with and without exogenous supplementation of glutathione (10 mM), as determined by CFU plating and counting. Each point represents two biological replicates, error bars indicate one standard deviation, and data are presented as mean ± SEM. Positive survival values indicate increases in CFU/mL. Measurements at 1× and 10× MIC (**a**) and endpoint measurements (**b**) across a range of antibiotic concentrations are shown. Arrows highlight protection. All MICs used were with respect to their baseline values, with the exception of increased concentration values adjusted for growth inhibition effects ("normalized"), as summarized in Supplementary Table 1. **c** Phase-contrast microscopy images of control and antibiotic-treated *E. coli* (10× MIC) with and without exogenous supplementation of glutathione (10 mM), taken ~10 h after continual antibiotic treatment. The starting densities of cells are similar across all images, and results are representative of two biological replicates. Scale bar, 3 μm. **d** Same as (**c**), but for treatment with peroxynitrite, a reactive metabolic by-product. Cells were imaged 3 h after treatment with peroxynitrite. Results are representative of two biological replicates. **e, f** Same as (**a, b**), but for peroxynitrite treatment and exogenous supplementation of glutathione (10 mM), dithiothreitol (10 mM), and mercaptoethanol (10 mM). **g** Same as (**a**), but for kanamycin and ciprofloxacin treatment and exogenous supplementation of dithiothreitol (10 mM), mercaptoethanol (10 mM), and α-tocopherol (50 mM). **h** Fractions of all cells that are condensed or lysed 6 h after antibiotic treatment (10× MIC) or peroxynitrite treatment (1 mM) and antioxidant pretreatment (glutathione, dithiothreitol, mercaptoethanol: 10 mM; α-tocopherol: 50 mM). Data from two different fields of view from two biological replicates; individual data points from fields of view are shown. The number of cells in each field of view in each group are indicated in parentheses, and p-values for two-sample t-tests for differences in mean value compared to cases with no antioxidant are shown next to cell numbers.

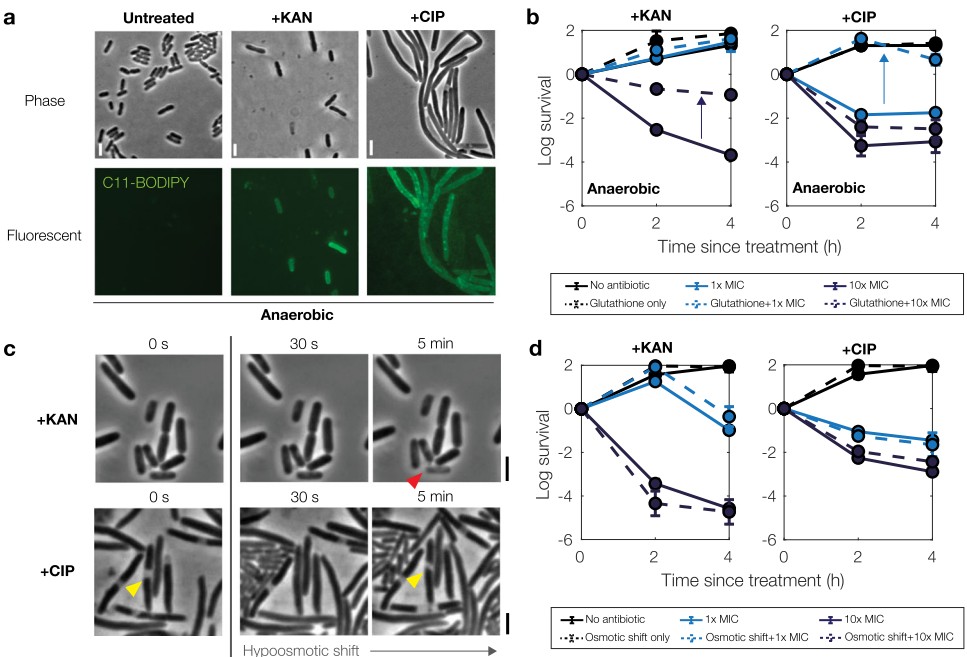

**Fig. 7 Cytoplasmic condensation and antibiotic killing are robust to changes in oxygen availability and external osmolarity. a** Phase-contrast and fluorescence microscopy images of control and antibiotic-treated *E. coli* (10× MIC) taken 3 h after treatment. Cells were grown, treated, and imaged under anaerobic conditions in LB medium. Scale bar, 3 μm. **b** Survival curves of *E. coli* under anaerobic kanamycin and ciprofloxacin treatment, as determined by CFU plating and counting. Each point represents two biological replicates, error bars indicate one standard deviation, and data are presented as mean ± SEM. Positive values indicate increases in CFU/mL. Arrows highlight protection. All MICs used were with respect to their baseline values under aerobic conditions (Supplementary Table 1). **c, d** Same as (**a, b**), but for aerobic cell cultures that are hypoosmotically shocked by downshifting from 250 mM sorbitol after ~3 h of antibiotic treatment (**c**) and at plating (**d**). Yellow and red markers highlight condensed and lysed cells, respectively.

In corresponding time-kill experiments, bulk cultures were treated with kanamycin or ciprofloxacin (10× MIC) in LB with 250 mM sorbitol. Cells were then hypoosmotically shocked by serial dilution in LB before plating on drug- and sorbitol-free LB agar. Consistent with our microscopy observations, subsequent CFU quantitation showed that cellular survival was not significantly changed in these cultures relative to the case of no osmotic shift at the time of plating (Fig. 7d). Together, these results suggest that physical changes in turgor pressure do not mitigate kanamycin- and ciprofloxacin-induced cell death and membrane damage. These findings are also consistent with the hypothesis that reactive metabolic by-product-mediated membrane damage lies upstream of cytoplasmic condensation and cell death.

## Discussion

In summary, our results, which include both single-cell biophysical and conventional bulk-culture approaches, suggest the following physical biology-based model of bacterial cell death. Upon the primary antibiotic–target interaction, cells generate reactive molecules from increased flux through metabolic pathways including glycolysis and the tricarboxylic acid cycle, as previously reported[3]. These reactive metabolic by-products lead to cellular damage by promiscuously reacting with nucleic acids, proteins, and membrane lipids. Reactions with the membrane in processes including lipid peroxidation induce loss of membrane integrity, which results in cytoplasmic condensation through the leakage of cytoplasmic contents and coincides with the loss of viability in cells. Intriguingly, because cytoplasmic condensation is not seen

in all dead cells, our results suggest that alternative cell death pathways—which may involve cellular damage to nucleic acids and proteins resulting from both the primary drug–target interaction and subsequent generation of reactive metabolic by-products—also contribute to antibiotic-induced cell death.

The phenotype of cytoplasmic condensation described in this work may offer a way of diagnosing bacterial cell death, in addition to suggesting possible molecular pathways that contribute to antibiotic lethality. Our observation that condensed cells no longer accumulate biomass at the time of condensation suggests, consistent with recent work from the Zhao and Drlica groups[11], that antibiotic-treated cells can die substantially after antibiotic treatment, and that this cell death is not revealed by traditional time-kill experiments in which cells are assumed to have died at the time of plating. As our observations indicate that condensed cells are effectively dead, screening for cytoplasmic condensation may inform a phenotype-based approach to antibiotic susceptibility testing, which typically involves cell culturing and genetic sequencing or intensive time-kill assays.

Finally, we observed that treatment by aminoglycoside and fluoroquinolone antibiotics, in contrast to their primary modes of action, is associated with membrane damage, lipid peroxidation, and changes in lipid composition. The observation that cytoplasmic condensation and lipid peroxidation can occur under different oxygen conditions, and are variably influenced by the application of different antioxidants, suggests that a complex landscape of reactive metabolic by-products may contribute to antibiotic lethality. Further work is needed to characterize these reactive metabolic by-products and the different cell death pathways in which they participate. The present study addresses this issue, in part, by showing a sequence of cellular events that can lead to antibiotic-induced cell death, underscoring cytoplasmic condensation as a terminal cell death phenotype in antibiotic-treated cells. Intriguingly, comparable cell death phenotypes have been observed in modes of killing as diverse as carbon starvation[42] and copper alloy treatment[43], suggesting that cytoplasmic condensation may be a common phenotype accompanying bacterial cell death.

## Methods

**Bacterial strains.** The wild-type strain of *E. coli* used in this study, unless other specified, is MG1655; we verified similar results in W3110 and BW25113. Strains with fluorescently tagged cytoplasmic and cell envelope components used in this study comprise those tagged as follows: cytoplasmic mCherry (DH5α(λpir), pTD47), ZipA-mCherry (inner membrane-tagged, TB28, attHKNP5)[44], LpoB-GFP (MG1655, attHKCB28; outer membrane-tagged)[44] and ssDsbA-LpoB-GFP (TB28, attHKCB41; periplasm-tagged)[45]; these strains were a gift from the Bernhardt group and cultured in the absence of antibiotics. The ΔgshA (JW2663-1) and Δgor (JW3467-1) strains were from the Keio collection of non-essential single knockouts[46]. For *B. subtilis*, strain 168 (ATCC 23857) was used.

**Antibiotics and other reagents.** Kanamycin sulfate (product 60615, MilliporeSigma, St. Louis, MO), gentamicin sulfate (MilliporeSigma G1914), mecillinam (MilliporeSigma 33447), and ampicillin sodium salt (MilliporeSigma A9518) were dissolved in ultrapure MilliQ-water. Ciprofloxacin powder (MilliporeSigma 17850) was dissolved in dilute acid (0.1 M HCl). Norfloxacin (MilliporeSigma N9890) was dissolved in dimethyl sulfoxide (DMSO, MilliporeSigma D5879). All antibiotics were freshly prepared before each experiment.

L-glutathione (reduced, MilliporeSigma G4251) and sorbitol were dissolved in ultrapure Milli-Q water. Solutions of DL-dithiothreitol (MilliporeSigma D0632) and 2-mercaptoethanol (MilliporeSigma M6250) were also prepared with ultrapure Milli-Q water. (+/-)-α-tocopherol (MilliporeSigma T3251) was dissolved in ethanol. Peroxynitrite (MilliporeSigma 516620) was used as supplied. All of these reagents were freshly prepared before each experiment.

**Bacterial culture and growth.** Cells were grown in liquid LB medium (product 244620, Becton Dickinson, Franklin Lakes, NJ). LB media containing 1.5% Difco agar (Becton Dickinson 244520) was used to grow individual colonies. Cells were grown from single colonies at 37 °C either in tubes in an incubation chamber shaking at 300 rpm or, for determining MICs, in 96-well plates (product 9018, Corning, Corning, NY) sealed with breathable membranes (MilliporeSigma

Z763624) with shaking at 900 rpm. For Supplementary Fig. 12, LB medium was diluted in phosphate-buffered saline (PBS; Corning 21-040).

**Determination of MICs.** We determined MICs for all antibiotics considered in this work by diluting both 1:100 and 1:10,000 from an overnight culture into both 96-well plates and 14-mL Falcon tubes with varying antibiotic concentrations. The MIC was determined as the minimum concentration at which no visible growth occurred overnight (12–18 h; $OD_{600} < 0.1$). $OD_{600}$ values were measured in 14 mL Falcon tubes with a Biowave cell density meter CO8000 (Biochrom, Holliston, MA) and in 96-well plates, using 200 μL working volumes, with a SpectraMax M3 plate reader (Molecular Devices, San Jose, CA). A summary of all MIC values thus determined and their working values for different bacterial strains are provided in Supplementary Table 1, and we note here that we found quantitatively similar MIC values between 1:100 and 1:10,000 dilutions, which lie within the tabulated ranges shown in Supplementary Table 1.

**Microscopy.** Microscopy experiments were performed with cells sandwiched between agarose pads and glass coverslips or slides unless otherwise stated. Cells were concentrated by centrifugation at 850$g$ for 5 min and resuspended in a smaller volume of supernatant. We placed 2 μL of the resuspended bacterial culture between either 3″ × 1″ × 1″ microscope slides (product 125444, Fisher Scientific, Hampton, NH) or number 1.5 coverslips (VWR, Radnor, PA) and 1 mm thick agarose (1.5%) pads made from growth media (agarose: type IX-A, MilliporeSigma A2576), following conventional procedures for immobilizing cells[47]. Cells were imaged immediately afterward. We used a Nikon Ti inverted microscope (Nikon, Tokyo, Japan) equipped with a 6.5 μm-pixel Hamamatsu CMOS camera (Hamamatsu, Hamamatsu City, Japan), a Nikon 100× NA 1.45 objective, TIRF illumination, and an enclosing custom-made incubation chamber and a Zeiss Axiovert 200 inverted microscope equipped with a Zeiss Axiocam 503 camera, a 40× objective (EC Plan-neofluar, NA 0.75), a 63× objective (EC Plan-neofluar, NA 1.25), and an enclosing custom-made incubation chamber for microfluidic experiments and fluorescence microscopy experiments. AFM experiments were performed with a Zeiss AxioObserver D1 (Zeiss, Jena, Germany), as mentioned below. For all other experiments and replicate fluorescence microscopy experiments, we used a Zeiss Axioscope A1 upright microscope equipped with a Zeiss Axiocam 503 camera, a Zeiss 100× NA 1.3 Plan-neofluar objective (Zeiss, Jena, Germany), and an X-Cite 120Q Iris FL light source (Excelitas Technologies, Salem, MA). For timelapse measurements, cells were imaged at 37 °C inside incubation chambers. Images were recorded using Zen Lite Blue (Zeiss), NIS-Elements (Nikon), and AxioVision (v.4.8, Zeiss) software. We used ImageJ (NIH, Bethesda, MD) for processing and analyzing timelapses and the StackReg plugin[48], which recursively aligns images in a sequence with geometric transformations, to correct for microscope drift as necessary. Epifluorescence exposure times were limited to a maximum of 300 ms to avoid photobleaching. All microscopy experiments were replicated at least twice, and we verified that the absolute values of all fluorescence intensities were comparable across experiments performed using the same microscopy setup.

**Microfluidics and osmotic shock experiments.** We used the CellASIC ONIX platform (MilliporeSigma, Burlington, MA) and the associated bacterial microfluidic plate (B04A-03). Cells were grown in culture to early log phase, optical density ($OD_{600}$) between 0.1 and 0.2, or stationary phase, $OD_{600}$ approximately 2, as described below, as measured in 14 mL Falcon tubes with a Biowave cell density meter CO8000. Cells were then loaded into the microfluidic chamber according to the manufacturer's instructions at a flow rate corresponding to 20 kPa (~10 μL/h). For antibiotic treatment, LB containing antibiotics was flowed in from two inlets at a flow rate corresponding to 20 kPa (~10 μL/h); under these settings, we verified that the solution saturated the device in approximately 5 min by adding yellow-green fluorescent beads (0.02 μm diameter; FluoSpheres F8760, Thermo-Fisher, Waltham, MA) to the solution as a proxy for its concentration. Cells were subjected to the same flow continuously unless otherwise noted. For wash-out experiments, fresh LB without antibiotics was flowed in from two inlets at a flow rate corresponding to 20 kPa (~10 μL/h) after cells were treated for specified durations by flow of antibiotic-containing LB. For the osmotic shock experiments shown in Fig. 1e, the protocol outlined above was followed with the following modifications. First, cells were cultured in the presence of antibiotic, when applicable, before being loaded into the microfluidic device. Second, after cell loading and equilibration, the flow was replaced by LB + antibiotic + sorbitol (500 mM; D-sorbitol, MilliporeSigma S1876) flowed in from two inlets at a flow rate corresponding to 20 kPa (~10 μL/h). For the osmotic shock experiments shown in Fig. 7c, d, *E. coli* W3110 was cultured in LB + sorbitol (250 mM). Cells were then loaded into the microfluidic device, and treated with constant flow of LB + sorbitol (250 mM) + antibiotic (when applicable) for 2–5 h until the onset of condensation. For washout experiments, the flow was then replaced by LB + antibiotic (when applicable) flowed in from two inlets at a flow rate corresponding to 10–20 kPa (~5 to 10 μL/h).

**Atomic force microscopy measurements.** A colony of *E. coli* MG1655 or W3110 was grown in LB media overnight at 30 °C with shaking at 220 rpm. A new culture

was then started from the overnight culture with a 1:1000 dilution in fresh LB media and grown at 37 °C with shaking at 220 rpm. At an $OD_{600}$ between 0.1 and 0.2, ciprofloxacin or kanamycin was added to the shaking culture to a final concentration of 10× MIC. After 3–4 h, 100 μL of bacterial solution was aliquoted on a glass-bottom petri dish (FluoroDish, WPI, Sarasota, FL) coated for 1 h in a 10 mM tris buffer solution (pH 8.5) containing 4 mg/mL dopamine hydrochloride (99%, MilliporeSigma H8502) and incubated at room temperature for 30 min. The bacterial cell solution was then carefully removed, and the petri dish was rinsed twice with PBS to remove unbound cells. PBS was then added to the petri dish for AFM measurements in liquid. AFM measurements were conducted using a Nanowizard IV (JPK BioAFM – Bruker Nano GmbH, Berlin, Germany) mounted on an inverted light microscope (AxioObserver D1, Zeiss). Commercial AFM tips with a nominal spring constant of 0.1 N/m (qp-BioAC, CB2, Nanosensors) were used. Images of immobilized bacteria were acquired using the Quantitative Imaging (QI$^{TM}$) mode of the instrument, a force-distance curve-based imaging mode which allowed us to simultaneously acquire several sample parameters (including topography and mechanical properties). The imaging parameter used were: 2 nN setpoint, 7 ms pixel time, 400 nm $z$-length. The manufacturer's (JPK BioAFM – Bruker Nano GmbH) data processing software was used to determine the elastic modulus from approach force-distance curves using the Hertz model modified for a paraboloid indenter[49]:

$$F = \frac{E}{1 - \nu^2} \frac{4\sqrt{R_C}}{3} \delta^{3/2}.$$

Here, $F$ is the applied force, $E$ is the elastic modulus, $\nu$ is the Poisson's ratio, $R_C$ is the radius of the tip, and $\delta$ is the indentation (vertical tip position). The cantilever was calibrated using routines within the AFM software based on the Sader method[50]. Representative force-distance curves, elastic modulus heatmaps, and cell dimension measurements are provided in Supplementary Fig. 7.

**Inference of the cellular turgor pressure from atomic force microscopy measurements**. Following seminal work by Arnoldi et al.[51], it is possible to estimate the turgor pressure of untreated, turgid cells from our AFM measurements. As shown in Supplementary Fig. 7, typical force-indentation curves on such cells indicate the effective spring constant of the AFM tip and the cell envelope as $k_{eff} \sim 0.03$ N/m. For an AFM tip with spring constant 0.1 N/m and assuming that the effective spring constant arises from two linear springs in series[51], the cell envelope spring constant is ~0.04 N/m. The following equation, based on modeling the indentation of a pressurized shell, then relates the cell envelope spring constant to the cellular turgor pressure:

$$k_s = \frac{3\pi}{2} pR\varphi(\rho/d). \tag{1}$$

Here, $k_s$ is the cell envelope spring constant, $p$ is the cellular turgor pressure, $R$ is the cell radius, and $\varphi(\rho/d)$ is a geometric factor, $\varphi(\rho/d) = \rho K_1(\rho/d)/[dK_0(\rho/d)]$, where $K_0$ and $K_1$ are modified Bessel functions, $\rho$ is the radius of the contact of the cantilever tip with the bacterial envelope, and $d$ is a characteristic annealing length of the cell envelope deformation[51]. For typical values of $R = 0.4$ μm and $\varphi(\rho/d) \approx 0.2$ relevant to our experiments, Eq. (1) indicates $p \sim 100$ kPa, an estimate quantitatively consistent with previous osmolarity-based studies of turgor pressure in *E. coli*[19] and approximately 3-fold larger than turgor pressure estimates derived from AFM experiments performed on bulging cells, for which cellular protrusions without the peptidoglycan cell wall were indented[52].

**Membrane permeability and depolarization assays**. SYTOX Blue Nucleic Acid Stain (product S11348, Invitrogen, Carlsbad, CA), a DNA intercalating dye, was added to incubating liquid cultures and agarose melt to a final concentration of 5 μM to stain cells with and without compromised membranes. DiBAC$_4$(3) (Invitrogen B438), a fluorescent reporter of membrane potential, was dissolved in DMSO and added to incubating liquid cultures and agarose melt to a final concentration of 10 μg/mL to stain cells with and without depolarized membranes. ION Potassium Green-2 AM (IPG; product ab142806, Abcam, Cambridge, UK), a membrane-permeable dye sensitive to potassium ions, was added to incubating liquid cultures and agarose melt to a final concentration of 40 μM to stain potassium ions in cells.

**ROS and RNS detection with fluorescent dyes**. For general detection of oxidative stress and ROS ($H_2O_2$, ROO·, and $ONOO^-$), we used the cell-permeant dye carboxy-$H_2$DCFDA (Invitrogen C400), which was dissolved in DMSO and added to incubating liquid cultures and agarose melt to a final concentration of 10 μM. For general detection of RNS, and in particular nitric oxide (NO), we used DAF-FM diacetate (Invitrogen D23844), dissolved in DMSO, at a final concentration of 10 μM.

**Membrane peroxidation detection with a fluorescent dye**. C11-BODIPY 581/591 (Invitrogen D3861), a fluorescent dye-based lipid peroxidation sensor whose fluorescence emission peak shifts from red to green upon lipid peroxidation[31], was dissolved in DMSO and added to incubating liquid cultures or agarose melt to a final concentration of 10 μM to stain membranes undergoing lipid peroxidation.

**Image analysis**. Cells were either counted manually or annotated semi-automatically using the MicrobeJ plugin[53] in ImageJ (below) to determine features, such as condensation and lysis, and cellular dimensions. Fluorescence was quantitatively analyzed with ImageJ (National Institutes of Health, Bethesda, MD). Briefly, closed cell contours were delineated based on phase-contrast images using MicrobeJ, and ImageJ was used to measure the mean fluorescence pixel intensity within a given contour. The average background fluorescence intensity was subtracted from all measured values, and the relative intensity was calculated by dividing the difference with the average background fluorescence value. Overlay images were generated with Adobe Photoshop (Adobe, San Jose, CA). We were not blinded to allocation in the analysis.

Condensed cells were delineated based on phase-contrast images or a combination of phase-contrast and fluorescence microscopy images, according to the features described in the main text and shown in Fig. 1a, b. The cell volumes shown in Fig. 2f were calculated, excluding cellular poles, by fitting 2D images of cells to bent rods with constant radii along midlines. Variations in cell length and radii across time were recorded using both phase-contrast timelapses and fluorescence timelapses from the cytoplasmic mCherry strain; see Supplementary Fig. 9 for representative traces. We further note that the reported frequencies of cytoplasmic condensation and lysis in Fig. 1d were determined by counting at predefined time intervals up to the end of imaging. It is possible that more cells are condensed or lysed than reported—this could occur if cells either became condensed or lysed after imaging or, in the case of condensation, if cells condensed and lysed faster than the time resolution (~1 min/frame) of our imaging.

**Single-cell traces**. Cell lengths were automatically measured in 20 min intervals for Fig. 3b, f and 10 min intervals for Fig. 3d (2 min intervals for the inset) using the MicrobeJ plugin in ImageJ. These measurements were repeated manually as a consistency check; we found that the results were in good quantitative agreement. Cell lengths were determined as the arclengths of cellular midlines, a construction which takes into account cell bending and movement across different frames of a timelapse[22]. A three-point moving average (four for the inset) was taken to smooth the curves in Fig. 3d using MATLAB (Mathworks, Natick, MA). Normalized growth rates were calculated by taking discrete finite differences of cell lengths.

**Sample preparation for thin-layer chromatography experiments**. A single colony of *E. coli* W3110 was grown overnight at 30 °C with shaking at 220 rpm. The next morning, a 1:200 dilution of the overnight culture in 100 mL of fresh LB was grown in Erlenmeyer flasks to an $OD_{600}$ of 0.22 at 37 °C with shaking at 220 rpm. At this $OD_{600}$, freshly prepared solutions of ampicillin (final concentration 100 μg/mL; 10× MIC), ciprofloxacin (final concentration 1 μg/mL; 10× MIC), kanamycin (final concentration 50 μg/mL; 10× MIC), and hydrogen peroxide (final concentration 10 mM; 4× the working MIC of 2.5 mM) were added to the cultures. A control (LB only) culture was examined in parallel. The cultures were incubated for 1, 3, and 6 h. At each timepoint, 20 mL samples from each culture were aliquoted, centrifuged at 3770g and 4 °C, and pelleted in a 50 mL Falcon tube. The supernatant was discarded and the pellet was resuspended in 1 mL of fresh LB, then added to a cryovial. The resuspended pellets were centrifuged again at 3770g and 4 °C, the supernatant was again discarded, and the pellet was flash-frozen in liquid nitrogen. The flash-frozen samples were then stored at −80 °C until further processing.

**Thin-layer chromatography of lipid content**. Frozen cell pellets were resuspended in 300 μL of ultrapure MilliQ-water and lipids were extracted in two steps with (i) 600 μL of chloroform-methanol (10:1, v/v), and (ii) 600 μL of chloroform-methanol (2:1, v/v). The lipid-containing organic phases were pooled and concentrated. Samples were volume-corrected so that each sample had identical amounts of phospholipid content which approximated the membrane lipid fraction. Samples were applied to 10 × 20 cm silica gel 60 HPTLC plates (Merck, Darmstadt, Germany) and chromatographically migrated using (i) chloroform-methanol-glacial acetic acid (65:25:10, v/v/v) over 50% of the running distance (3.5 cm), and (ii) *n*-hexane-diethylether-glacial acetic acid (70:30:1, v/v/v) over 100% of the running distance (7 cm). The lipids were stained with solutions of copper (II) sulfate-orthophosphoric acid or primuline. Peak area intensity measurements were performed on TLC images using ImageJ. The assay was repeated twice.

**Time-kill assays and CFU measurements**. For all time-kill assays in bulk culture, cells were diluted 1:100 from an overnight culture into 14 mL Falcon tubes containing 2 mL of growth media and incubated either to early log phase ($OD_{600} \approx 0.1$), to early stationary phase ($OD_{600} = 1.0–1.5$), or to late stationary phase ($OD_{600} > 2.0$) in the conditions described above (with shaking at 300 rpm at 37 °C). $OD_{600}$ measurements were taken in 14 mL Falcon tubes with a Biowave cell density meter CO8000 and in 96-well plates, using 300 μL working volumes, with a SpectraMax M3 plate reader. For antioxidant pretreatment, antioxidants were added to growth media 10 min before the addition of antibiotics, when relevant. Antibiotics were added to the final concentrations indicated, and cultures were re-incubated with shaking at 300 rpm at 37 °C. At the indicated times, cells were removed from incubation, aliquoted, and serially diluted in LB and spotted or spread on LB agar. We performed serial dilutions in LB instead of other media, like PBS, in order to

better control for osmolarity and nutrient shifts. For antibiotic concentrations at $10\times$ MIC, cells were also centrifuged at $2350g$ for 1 min, 90% of the supernatant was removed, and cells were resuspended in a smaller volume before being serially diluted so as to aid in the removal of antibiotics. Petri dishes were allowed to dry at room temperature before incubation at 37 °C overnight (16–24 h). CFUs were then determined by manual counting, and all measurements are based on counts containing at least 10 colonies. Survival was determined by normalizing all CFU/mL measurements to that immediately before antibiotic treatment at time 0 h.

**Flow cytometry.** Cells were grown and treated with antibiotics as described above. At the indicated times, cells grown in the presence of fluorescent dyes were removed from incubation, aliquoted into 96-well plates, and processed with a CytoFLEX S flow cytometer (Beckman Coulter, Brea, CA). Cells were flowed at a low flow rate (10 μL/min), and 20,000 events were recorded for each sample. Cells were filtered by calibration of forward- and side-scattering measurements based on those of the cytoplasmic mCherry strain, which differs from MG1655 in that its red fluorescence can be used as a ground truth for separating cells from debris (Supplementary Fig. 16). All samples were analyzed at least in biological duplicate. FlowJo software (v10, Becton Dickinson) was used for data post-processing, and fluorescence intensities correspond to those derived from pulse areas.

**ATP abundance assay.** Intracellular ATP (Supplementary Fig. 12) was quantified using the BacTiter-Glo Microbial Cell Viability Assay (product G8230, Promega, Madison, WI), according to the manufacturer's instructions. Luminescence and absorbance of samples were measured, using 200 μL working volumes, with a SpectraMax M3 plate reader.

**Anaerobic chamber.** The experiments shown in Fig. 7a, b were performed in an anaerobic chamber (Type B Vinyl, Coy Labs, Grass Lake, MI) equipped with twin palladium catalysts and a Coy Oxygen/Hydrogen Analyzer (Coy Labs) and maintained at 37 °C. A 5% hydrogen in nitrogen gas mix (product NI HY5300, AirGas, Radnor, PA) was used to maintain the steady-state anaerobic environment at less than 5 ppm oxygen. Additionally, a BD BBL GasPak anaerobic indicator (Becton Dickinson), and growth media containing 1 mg/L resorufin (resazurin: MilliporeSigma R7017) as an anaerobic indicator[54], were used to ensure strictly anaerobic conditions. All starting cultures were taken from cultures grown overnight inside the anaerobic chamber, and MICs were based on their aerobic values.

For time-kill assays, cells were serially diluted, plated, and grown overnight inside the anaerobic chamber. Cell cultures were treated in early log phase, $OD_{600} \approx 0.02$. $OD_{600}$ measurements were taken in 96-well plates, using 300 μL working volumes, with a SpectraMax M3 plate reader. For microscopy experiments, cells were plated and imaged with a Zeiss Axioscope A1 upright microscope, as detailed above; this microscope was used because it could fit inside the anaerobic chamber. Plating and microscopy were performed strictly inside the anaerobic chamber. To ensure strictly anaerobic conditions, cells were not immobilized with LB agarose pads when imaging in the anaerobic chamber, as compared to imaging under aerobic conditions.

To allow for deoxygenation, all materials used in our experiments were brought into the anaerobic chamber at least 24 h before the start of each experiment, with the exception of antibiotics, C11-BODIPY dye, and glutathione. To ensure freshness of these reagents, these reagents were prepared immediately before each experiment, brought into the anaerobic chamber, and equilibrated in open-cap 1.5 mL or 5 mL centrifuge tubes for 1 to 2 h before usage. After the addition of these reagents to growing cultures, we observed that resorufin in the growth media remained strictly colorless, indicating that these reagents did not introduce significant sources of environmental oxygen to our experiments.

**Protein concentration and LPS level assays.** The LPS levels shown in Supplementary Fig. 13 were assayed as follows. Here, 50 mL of log-phase bulk cultures of *E. coli* MG1655 grown in LB were treated with kanamycin, ciprofloxacin, and ampicillin at $10\times$ MIC for 3 h in 250-mL flasks. Cells were then centrifuged at $3720g$ for 10 min. Next, the supernatant in all samples was discarded. Each cell pellet was washed with 500 μL PBS, resuspended, and centrifuged again as above; then, the supernatant was discarded from each sample. Next, 500 μL B-PER II (product 78260, Thermo Fisher Scientific, Waltham, MA) containing 100 μg/mL lysozyme (MilliporeSigma L6876) and 5 U/mL DNase I (Thermo Fisher 90083) was added to cell pellets for harvesting, and all samples were vortexed and incubated at 37 °C for an additional 15 min. All samples were then centrifuged at $1500g$ for 10 min, and the supernatant was aliquoted from each sample for analysis. Serial 1:10, 1:100, 1:10,000, and 1:100,000 dilutions of each cell lysate sample in PBS and endotoxin-free water were generated and used for each of the assays below to ensure that protein and LPS levels were in the range of the standard curves.

To measure the total protein concentration of cell lysates, we used a Pierce BCA protein assay kit (Thermo Fisher 23227) following the manufacturer's instructions. Briefly, the assay reagent was prepared by mixing the assay kit components according to the manufacturer's instructions, and, for each sample, 25 μL of the cell lysate was pipetted directly into a larger volume (200 μL) of reagent in a 96-well plate. All samples were then incubated for 30 min at 37 °C according to the manufacturer's instructions, and the optical density at 562 nm was measured with a

SpectraMax M3 plate reader. Standard curves were generated from control samples with bovine serum albumin (BSA) concentrations of 2000, 1500, 1000, 750, 500, 250, 125, 25, and 0 μg/mL, according to the manufacturer's instructions. To determine the component of protein concentration originating from bacterial cells, the protein concentration of the B-PER II medium with 100 μg/mL lysozyme and 5 U/mL DNase I (measured to be ~140 μg/mL) was subtracted from the calculated protein concentrations in each lysate sample.

To measure the LPS levels of cell lysates, we used a Pierce Chromogenic Endotoxin Quant Kit (Thermo Fisher A39553) following the manufacturer's instructions. Briefly, the assay reagent, based on the LAL, was reconstituted and applied to 50 μL of samples and LPS standards in a 96-well plate. After mixing and incubation, chromogenic substrate was added. Following a final incubation step and addition of 25% acetic acid to each well, the optical density at 405 nm was measured with a SpectraMax M3 plate reader. Standard curves were generated from control samples with LPS concentrations of 1.0, 0.5, 0.25, 0.1, and 0 endotoxin units per mL, according to the manufacturer's instructions.

**Statistical testing.** Two-sample Kolmogorov-Smirnov tests and one-sample and two-sample *t*-tests were performed at the standard 5% significance level, and *p*-values are indicated when applicable.

**Reporting summary.** Further information on research design is available in the Nature Research Reporting Summary linked to this article.

## Data availability
Additional data that support the findings of this study are available from the corresponding authors upon reasonable request. Source data are provided with this paper.

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

## Acknowledgements

We thank Tom Bernhardt and Anastasiya Yakhnina for assorted *E. coli* strains, Ünal Coskun and Michal Grzybek for assistance with lipid analyses, and Allison J. Lopatkin, Ian W. Andrews, Emma J. Chory, Erica J. Zheng, and members of the Collins lab for helpful discussions and comments. F.W. was supported by the James S. McDonnell Foundation. J.M.S. was supported by the Banting Fellowships Program (393360). L.D.R. was supported by the Volkswagen Foundation. J.J.C. was supported by the Defense Threat Reduction Agency (grant number HDTRA1-15-1-0051), the National Institutes of Health (grant number R01-AI146194), and the Broad Institute of MIT and Harvard.

## Author contributions

F.W. conceived research, performed experiments and analysis, developed the model, wrote the paper, and supervised research. J.M.S. provided assistance with data interpretation and manuscript editing. B.C. provided assistance with experiments. S.P. performed experiments. J.F. and L.D.R. conceived research and performed experiments and analysis. J.J.C. conceived and supervised research.

## Competing interests

J.J.C. is scientific co-founder and scientific advisory board chair of EnBiotix, an antibiotic drug discovery company. The remaining authors declare no competing interests.
