## [Peer Review File · Nature Communications]

REVIEWER COMMENTS

Reviewer #1 (Remarks to the Author):

The manuscript (NCOMMS-20-36348) by Wong et al. reported that cytoplasmic condensation induced by ROS/RNS-mediated membrane damage correlates with killing by fluoroquinolone and aminoglycoside antibiotics. It is now widely accepted that ROS, RNS, or other types reactive species generated during metabolic process play a casual role in antibiotic-mediated killing. However, how these reactive species specifically kill bacteria in downstream events remain poorly understood. The present work used comprehensive bacteriological, biochemical, and biophysical approaches to show that cytoplasmic condensation correlates with ROS accumulation, membrane damage, and cell death. The present study provides a step forward along this line of work.

My major issue with the work is that co-occurrence or correlation with antimicrobial killing would be much less interesting than a causal role, but the authors failed to distinguish these possibilities in their work. From my examination of their data, condensation seems to occur after cell death because it occurs long after the majority of cells have been killed by antibiotic treatment: at 10 X MIC, kanamycin and ciprofloxacin both kill more than 4-6 logs of *E. coli* cells grown in LB medium. Thus, cytoplasmic condensation is probably an intermediate, co-occidental/consequential event of ROS-mediated membrane damage of dead cells and the subsequent lysis of these cells. The authors' own observation that condensed cells failed to recover after antibiotic removal further supports the above conclusion. Thus, unless the authors can demonstrate what they observed is not from cells that have already long been killed by the antibiotics, the finding of an intermediate marker of dead, decaying cells before lysis becomes much less interesting. The work would be stronger if the authors can perform at least one experiment that validates a causal role of condensation to death. A simple clarification experiment is to use an isotonic medium to perform the killing and cytosolic condensation experiments to see whether killing is maintained while condensation and lysis are eliminated/reduced/delayed. A positive result would argue against involvement of condensation in cell death.

Minor issues

1. Lines 63-68: What percentage of total cells showed cytoplasmic condensation at the time of condensation measurement? If the percentage is very small, then condensation, even as only a phenotypic marker for cell death, is not very useful because it would drastically underestimate dead cells. It is also well known that both kanamycin and ciprofloxacin kill *E. coli* cells rapidly and extensively without lysing them. Is this the reason why the condensation and lysis experiments were done after 3 h of antibiotic treatment, e.g. long after the majority of cells had been killed?
2. Lines 65 and 72: If condensation occurs within one minute, how long does the phenotype persist before lysis occurs? Rapid loss of condensation may explain why only a small fraction of cells could be captured to show the condensation phenotype.
3. Lines 97-102: Why did the increase in SYTOX Blue fluorescence occur at different stages with kanamycin- and ciprofloxacin-treated cells? Does condensation progress into lysis much quicker with kanamycin-treated cells than with ciprofloxacin-treated cells?
4. Lines 135-137 and 142-145: Since cells showing condensation failed to elongate or even filament and since filamentous cells, but not condensed cells, could resume growth after antibiotic removal, condensation must have occurred after the irreversible death step is complete. If so, what is the relevance of condensation to the death process?
5. Lines 164-165: How could high ROS fluorescence signals be observed with lysis and condensed cells? The membrane integrity of these cells had already been lost, ROS production should have been stopped, and the ROS-sensitive dyes would have readily leaked out of the broken cells.
6. Lines 187-190: Does glutathione affect MIC of kanamycin or ciprofloxacin? If it does, killing should be done using concentrations normalized to a certain fold of MIC rather than using the same absolute antibiotic concentration. This is important for obtaining a valid comparison.
7. Line 209: The word "many" is a value judgement that needs to be quantitatively supported. What is the percentage of dead cells that exhibit the condensation phenotype?
8. Legends of many figures are difficult for general readers to follow. The figures themselves should be easily understood without reference to the Methods or Results sections.

Reviewer #2 (Remarks to the Author):

In this article, Wong et al investigate the impact of aminoglycosides and fluoroquinolones on *E. coli* at the single cell level. They report that these two classes of antibiotics indirectly damage the membrane via toxic by-products, which causes the outflow of the cytoplasmic content and cytoplasm condensation, which coincides with loss of cell viability. The authors also report that pre-treating the cells with the antioxidant GSH attenuates cell death. The main message of the paper will be of general interest to the microbiology community and to people active in antibiotic research, in part by suggesting lipid peroxidation as a valid target for antibiotic development. Although I find the work interesting, I have a number of concerns that need to be addressed; the paper currently lacks key control experiments and crucial data.

1. Cytoplasmic condensation is defined by the authors as regions growing denser; in Fig S2, they used as read-out the density of phase contrast measurements on 18 cells. Another method should be used to better assess cytosolic packing, for instance using a cytoplasmic fluorescent probe like mCherry.

2. The number of cells analyzed and the methodology used (manual detection) are far from being optimal. I do not agree with the authors when they justify (lines 438-446) that "pre-existing software, such as CurvatureTracker or MicrobeTracker, did not provide adequate tools for automating the collection of data required". These tools are in fact versatile and able to detect very complex shapes. If the authors experienced difficulties with these software, they should try a tool like microbeJ which is more user-friendly. Using these tools is important to allow an unbiased and statistically-relevant analysis of microscopic images. Manual detection of the cell edges can lead to approximation, especially if the measurements have been performed as in fig.1e by simply "drawing a straight line" between the two poles without taking into account the bending of the cell and its movement in response to the modification of the composition and flow of the liquid in the chamber of the microfluidics device.

3. Line 65: the sentence "discrete portions of the cytoplasm typically became phase-light" should be rephrased: these phase-light zones correspond to an increased periplasmic space.

4. Line 70-72: It is unclear if the cytoplasm separates into different sub-compartments which are not connected to each other or not. Could the authors comment on that? Also, did the authors test the reversibility of the plasmolysis they observe?

5. Lines 81-95: turgor pressure is caused by an osmotic flow of water. A loss of turgor pressure induces an exit of water from a compartment of lower solute concentration into a compartment of higher solute concentration. Is the extra-cytoplasmic environment of the condensed cells hyper osmotic?

6. The authors propose that reactive by-products cause membrane damage, then cytoplasmic solutes go out, and finally the cytoplasm becomes condensed. But, what is the evidence that solutes go out? This should be tested using small cytoplasmic fluorescent proteins for instance. Further experiments are required to address this and clarify the mechanism.

7. Why do stationary-phase cells show less cytoplasmic condensation (lines 149-151)? Were non-condensed treated cells in exponential phase not elongating? Does a slower growth rate negatively impact cytoplasmic condensation? This needs to be discussed.

8. Membrane damage is crucial to explain the cause of death. Therefore, the authors need to test the validity of the various dyes they used as a proxy for membrane damage using normal plasmolysed cells: these cells should have no membrane damage.

9. Clearly establishing lipid peroxidation is also very important to validate the mechanism. The observation made using the fluorescence probe is indirect. A lipidomic study should be done to characterize the membrane damage that occurs and causes the leak. Also, the authors should test

the impact of carrying out the experiments anaerobically to see what happens in the absence of oxygen. Did the authors try adding fatty acid to rescue the condensed cells?

10. Line173: A positive control is missing: what is the impact of adding an external source of peroxyxynitrite to the untreated cells? Does combining peroxyxynitrite and either Kan or CIP further increase the number of condensed/lysed cells?

11. The authors should test whether gor or gshA mutants are more sensitive to antibiotic treatment. Also, did they try using other antioxidants, like DTT or mercaptoethanol? Is the lethality of the peroxyxynitrite-treated cells rescued by the addition of glutathione? Also, in Fig 4i (with KAN), representative images do not suggest that "At the single-cell level, glutathione largely alleviated condensation and lysis across a range of treatment conditions"

12-The mode of death of the antibiotic-treated cells is reminiscent of that of cells with a dominant mutation in MlaA (MlaA*; Sutterlin et al, 2016, PNAS). In the 2016 study, the authors argue that these phenotypes result from an increase of LPS in the outer membrane and from a futile flow of lipids from the inner to the outer membrane. Did the authors consider a loss of lipids from the inner membrane and did they check the LPS levels?

13. The author should discuss the death mechanism of the fraction of the population that does not display cytoplasmic condensation.

14. Fig. S3: the quality of the fluorescence signal is poor and not interpretable.

Reviewer #3 (Remarks to the Author):

The manuscript by Félix Wong et al. describes the study of how antibiotics induce cytoplasmic condensation through membrane damage and subsequent outflow of cytoplasmic contents. Topic and relevance of results match Nature Communications very well.

The manuscript is well written although highly condensed at some places more in the Nature style. For Nat. Communications, as the space limit is not so strict, I would strongly suggest to the authors to explain more in details the experiments and the results to facilitate its reading.

-Lines49-61: Why is the duration of treatment different between AMP (1h), CIP (6h) and KAN (3h)?

-When looking at Fig 1a and in particular to the "untreated cells" (see Zoom#1; attached), it seems obvious that some of the untreated cells are also lysed (cells a much less dark & with some darker parts localized at the poles or in the septum area). This means that these effects cannot only be exclusively attributed to the antibiotics itself. Surprisingly in Fig 1d, for the untreated cells the population fraction of lysed cells remains at "0". How do you explain this?

As a control the untreated cells should also be labelled with the fluorescent markers as in Fig 1b.

-Fig 1d. Is this quantification performed based on the phase contrast image? As shown in Fig 1b, for some cells (last row) the condensation on the phase contrast image is not evident, but on the contrary much easier to observe on the fluorescence image. Therefore this quantification would be more accurate if done on the fluorescence images.

-Fig1f: The authors analyzed by AFM the elastic modulus of cells treated or not with CIP. What is the applied force used to probe the elastic modulus? Which model did they used? Could the authors provide in SI some force-indentation curves with the fitted model? How was calibrated the spring constant of the tip? As the authors used the QI mode, they should also have access to the simultaneously recorded elasticity map showing the distribution of the Young modulus. This should also be presented to further evidence the localization of the different part of the cell with altered elastic properties. Are the results similar for the cell treated with KAN?

-The size of the cell before and after treatment with antibiotics was studied by light microscopy.

However, the resolution of such a method is quite low. As the authors have access to an AFM, it would be interesting to analyze the length and height (diameter) of cells (untreated or treated with CIP and KAN) by AFM. It would be interesting to see whether or not the decrease in length is associated with a change in diameter.

-Line 89-98. From the hyperosmotic shocks experiments, the authors conclude that the turgor of condensed and lysed cells is significantly diminished compared to turgid cells. However the AFM experiments suggest the opposite. As an increase in the elastic modulus is observed, it means that the cells are more rigid suggesting an increase in the turgor pressure. This should be clarified. The authors could use biophysical model to extract the turgor pressure from AFM indentation experiments performed on the bacteria cells?

-The biophysical model is very interesting. What is the evidences of the nanometer scale membrane defects? An alternative model could be that the condensation of the cytoplasm could lead to an increase in the water activity and therefore water outflow leading to a decrease in the cell volume. By comparison, this is something observed for mammalian cells just before mitosis, during pre-mitotic condensation. After the chromatin condensation, the cell volume decreases. This process involves the outward directed movement of chloride followed by water.

-In figure 2f, the model is compared to the empirically observed decreases in cellular volume. How do you measure the volume experimentally? Since you observed a 20% decrease in length, do you assume that the radius remains constant? This must be validated by AFM experiments to have precise estimate of the radius before and after treatment. It would also be useful to validate the biophysical model.

Reviewer #4 (Remarks to the Author):

Wong et al. present an interesting single cell based study that tries to elucidate the finale steps in the cascade of the mechanism of action of specific groups of antibiotics that yield a microscopically observable phenotype of a condensed cytoplasm. The manuscript is well structured, well written, the experiments build on each other and the conclusions are in general sound. The model describes cell death as a sequence of events in which antibiotic interaction with its main substrate creates metabolic byproducts that damage the fatty acids of the cell membrane and ultimately cause the creation of holes in the cell envelope. Outflow of cytoplasm through these holes results in a change of turgor pressure and eventually to condensation of the cytoplasm, an irreversible cell death phenotype.

I like the manuscript in general, especially the variety of targeted single cell experiments combined with rigorous biophysical modeling. To my knowledge, the major point of the paper addressing drug induced cytoplasmic condensation is novel. The Methods are detailed and well written. I would thus recommend publication after the following concerns/questions have been addressed:

- The authors argue that metabolic byproducts (free oxygen radicals) may cause lipid peroxidation that results in rupture of the membrane. Their experiments seem to support this hypothesis, however, this still leaves the question how a hole forms in the cell wall. Or can the cell wall be considered porous enough for the cytoplasm to flow through basically unhindered?

- Similarly, the mathematical model assumes that straight holes are formed that extend through all three layers of the cell envelope. I am guessing this is just a simplification for modeling purposes as Fig. 1c depicts holes in the inner and outer membrane that do not overlap?

Independent of that, the picture seems to be that static holes are being created in the cell envelope. I would argue that membrane fluidity rapidly closes these holes (~1 sec). I think cells undergoing electroporation or chemical transformation face a similar situation: small holes are being created such that DNA can pass through, however, the cytoplasm does not condense, and most cells survive. Is the idea behind the model that there are many holes that open and close constantly? I don't think this is reflected in the mathematical model. Could the authors please elaborate?

- The mathematical model to describe the temporal outflow of solute seems sound, justified by the

excellent agreement with experimental data shown in Fig. 2f and consistency checks shown in the Supplement. I think, however, some of the simplifications used to simplify the math should be rationalized better. Firstly, the authors assume that there is no water in the periplasm, which I think is not entirely correct. The water content of the periplasm also seems to be dependent on the turgor pressure (Sochaki 2011 Biophysical Journal - Protein Diffusion in the Periplasm of E. coli under Osmotic Stress).

- The mathematical model further assumes that all three layers of the cell envelope are disconnected. However, there are many protein complexes such as secretins that span the entire envelope and basically pin them together. Similarly, the model doesn't include the load bearing effect of cytoskeletal elements. Are these negligible?

- I have trouble with some of the statistical testing. For example, Fig. 4a,b,c: The results for untreated and turgid are mostly marked as * or ** significance, but the plots shown don't seem to reflect that. Similarly and looking at Supplementary Table 3, it seems that both means of the respective comparisons are identical within much less than one standard deviation. Could the authors please explain? I think it might make sense to show the full distributions in scatter plots, maybe as additional Supplement.

Minor points:

- The inclusion of ampicillin seems a bit confusing since it is not actually studied. Is this meant to be a sort of control for an antibiotic that does not induce cytoplasmic condensation?

- I like the timelapse plots (Fig. 1i, Fig. 4d) that correlate cytoplasmic condensation with membrane damage, depolymerization, and lipid oxidation. However, the presented intervals of 10 min make it hard to differentiate correlation and causation, especially since the experiments in Fig. 1d and results of the mathematical model suggest that these things happen on the minute or even seconds time scale. I'd love to see at least one plot with a much improved time resolution.

- Although the authors report that the difference in elastic modulus between untreated cells and the phase light region of treated cells is not significant, it still seems that phase light regions are at least as stiff as untreated cells. I would have expected a significant decrease. Can the authors explain?

- In Fig. 1b, the outer membrane label looks more like a cytoplasmic label, especially compared to the inner membrane label that seems to localize much better to the cell envelope. How is this possible?

- In line 113 the authors introduce their mathematical model without any explanation of what exactly is being model. I think the authors did a great job laying out the model in the Supplement and I encourage them to add a brief summary (2-3 sentences) in the main text.

- Fig. 3 b,d,f: I think it would be interesting to see/mark when the drug was added as well.

- The methods concerning microfluidics mention the flow rate in units of Pascal, which seems to be a device setting? Could the authors specify either a flow through in volume per time and channel dimensions and/or a shear rate?

Respectfully,
Matthias Koch

Response to reviewers

MS# NCOMMS-20-36348

Wong, Stokes, Cervantes, Penkov, Friedrichs, Renner, and Collins, "Cytoplasmic condensation induced by membrane damage is associated with antibiotic lethality"

Summary of main changes.

We thank all the reviewers for their constructive and thoughtful comments on the paper. We have addressed all of the points raised by the reviewers through additional experiments, analyses, and revisions, which have significantly strengthened the work. We would like to highlight the following key additions and revisions to the paper:

(1) To address Reviewer #1's request for a clarification experiment using isotonic medium, we have performed additional time-kill and microfluidic experiments to probe the effects of hypoosmotic shocks on cytoplasmic condensation and cell death. Consistent with our finding that cytoplasmic condensation is induced by membrane damage upstream of cellular osmolarity changes, we found that application of hypoosmotic shock did not significantly change cytoplasmic condensation or antibiotic lethality.

(2) As suggested by Reviewer #2, we have performed additional experiments and analyses that: (1) support previous biophysical measurements of the condensation phenotype; (2) clarify the changes in lipid composition associated with membrane damage; (3) provide positive controls for treatment with peroxyntirite; (4) provide additional measurements of LPS levels; and (5) provide data on other perturbations including different antioxidants, anaerobic conditions, genetic knockouts, and osmolarity conditions.

(3) As suggested by Reviewer #3, additional details regarding the AFM measurements have been provided in the main text, Methods, and Supplementary Information. Furthermore, the main text has been expanded to explain our experiments and results in greater detail.

(4) As suggested by Reviewer #4, we have better clarified aspects of the biophysical model in the main text and Supplementary Information.

In the following, line numbers and citations refer to the revised version of the paper, and responses are indicated in blue font.

Reviewer #1.

We thank the reviewer for their interest and insightful comments.

- 1. The manuscript (NCOMMS-20-36348) by Wong et al. reported that cytoplasmic condensation induced by ROS/RNS-mediated membrane damage correlates with killing by fluoroquinolone and aminoglycoside antibiotics. It is now widely accepted that ROS, RNS, or other types reactive species generated during metabolic process play a casual role in antibiotic-mediated killing. However, how these reactive species specifically kill bacteria in downstream events remain poorly understood. The present work used comprehensive bacteriological, biochemical, and biophysical approaches to show that cytoplasmic condensation correlates with ROS accumulation, membrane damage, and cell death. The present study provides a step forward along this line of work.**

We thank the reviewer for their appreciation of our work and hope that our revisions and responses provided below sufficiently address their comments.

- 2. My major issue with the work is that co-incidence or correlation with antimicrobial killing would be much less interesting than a causal role, but the authors failed to distinguish these possibilities in their work. From my examination of their data, condensation seems to occur after cell death because it**

occurs long after the majority of cells have been killed by antibiotic treatment: at 10 X MIC, kanamycin and ciprofloxacin both kill more than 4-6 logs of *E. coli* cells grown in LB medium. Thus, cytoplasmic condensation is probably an intermediate, co-incident/consequential event of ROS-mediated membrane damage of dead cells and the subsequent lysis of these cells. The authors' own observation that condensed cells failed to recover after antibiotic removal further supports the above conclusion.

The reviewer is correct that kanamycin and ciprofloxacin both kill several logs of *E. coli* cells grown in LB. We would like to point out that this killing is assayed through traditional microbiological time-kill assays, which involve plating on LB agar and overnight incubation. As such, the condensation phenotype we observe minutes to hours after antibiotic treatment may well occur before, or coincide with, cell death when cell death occurs after plating (see also response #4 below). Consistent with this possibility, our long-term microfluidic experiments show that cells accumulate biomass up to the point of condensation, several hours after application of antibiotics (Fig. 3 and Supplementary Movies 1-3). Regardless, we agree that the production of reactive metabolic byproducts upstream of cytoplasmic condensation and membrane damage contributes to cell death by damaging other cellular components, as ongoing work from our lab and others has shown, in addition to membrane lipids. The present study extends this hypothesis by focusing specifically on how the oxidation of membrane lipids could induce cell death.

- 3. Thus, unless the authors can demonstrate what they observed is not from cells that have already long been killed by the antibiotics, the finding of an intermediate marker of dead, decaying cells before lysis becomes much less interesting. The work would be stronger if the authors can perform at least one experiment that validates a causal role of condensation to death. A simple clarification experiment is to use an isotonic medium to perform the killing and cytosolic condensation experiments to see whether killing is maintained while condensation and lysis are eliminated/reduced/delayed. A positive result would argue against involvement of condensation in cell death.**

We agree that the work could be strengthened by showing a causal role of condensation to death. Our attempts to address this include performing (1) additional time-kill and microfluidic experiments to probe the effects of isotonic media on cytoplasmic condensation and cell death, as suggested by the reviewer; and (2) additional time-kill experiments with the application of other antioxidants that may scavenge reactive metabolic byproducts or mitigate lipid peroxidation, with the aim of rescuing antibiotic-treated cells.

First, consistent with our finding that cytoplasmic condensation is induced by membrane damage upstream of cellular osmolarity changes, we found that application of a 250 mM hypoosmotic shock did not significantly change cytoplasmic condensation or antibiotic lethality, as summarized beginning on page 16. We believe that this result reinforces the connection of cytoplasmic condensation to upstream reactive metabolic byproduct-mediated events, suggesting that it is not cytoplasmic condensation but rather membrane damage, possibly induced by lipid peroxidation, that contributes to cell death. Indeed, we point out that application of the antioxidant glutathione suppresses membrane damage, cytoplasmic condensation, and cell death (Fig. 6a-c,h), ostensibly by scavenging reactive metabolic byproducts upstream of these processes.

Second, our attempts at rescuing antibiotic-treated cells by (1) exogenous supplementation of dithiothreitol and mercaptoethanol, two other antioxidants, and (2) exogenous supplementation of the lipophilic antioxidant α -tocopherol, the latter of which have been evidenced to mitigate lipid peroxidation in eukaryotes, did not alter antibiotic lethality. Consistent with our finding that cytoplasmic condensation is associated with cell death, these two approaches also did not alter the emergence of cytoplasmic condensation in antibiotic-treated cells. We speculate that these two methods of scavenging reactive metabolic byproducts and mitigating lipid peroxidation may be less potent in *E. coli* than in eukaryotes; consistent with this idea, microscopy measurements showed that C11-BODIPY still exhibited substantial fluorescence in cells treated by exogenous α -tocopherol (Supplementary Fig. 15). Intriguingly, however, we note that recent work (ref. 38) applying an α -tocopherol analogue has reported decreased ciprofloxacin killing and alleviated lipid peroxidation in *E. coli*. We have pointed out this work and summarized our additional findings beginning on line 327 of the main text.

In sum, we believe that it would be intriguing to show that the membrane damage we have uncovered is the cause of antibiotic lethality. However, as suggested by the additional experiments we have performed, doing so appears to require a way to exogenously repair damaged cell membranes. To our knowledge, there are no experimental methods to do so, and the most relevant process that may be targeted, based on our observations, appears to be lipid peroxidation. We have therefore revised the text following the reviewer's important comment to include the additional experiments mentioned above and to focus on the accompanying—and, as we carefully distinguish, not necessarily causal—manifestation of cytoplasmic condensation in cell death.

- 4. Lines 63-68: What percentage of total cells showed cytoplasmic condensation at the time of condensation measurement? If the percentage is very small, then condensation, even as only a phenotypic marker for cell death, is not very useful because it would drastically underestimate dead cells. It is also well known that both kanamycin and ciprofloxacin kill E. coli cells rapidly and extensively without lysing them. Is this the reason why the condensation and lysis experiments were done after 3 h of antibiotic treatment, e.g. long after the majority of cells had been killed?**

We thank the reviewer for this comment and wish to point out that, for frequency measurements, we have measured condensation (and lysis) across time and not at a specific time (Fig. 1d). The frequency of condensation and lysis at a certain timepoint depends on when cells are observed, as well as the antibiotic concentrations applied, as shown in Supplementary Movies 1-3: such frequencies can be as large as ~90% of a field of view depending on these two factors, as shown in Supplementary Movie 3 for condensation in the case of high concentrations of ciprofloxacin. In contrast, antibiotic-tolerant stationary-phase cells, or glutathione-pretreated cells, treated with kanamycin or ciprofloxacin display noticeably less, or no, condensation and lysis compared to cells treated at the same antibiotic concentrations (Fig. 6c and Supplementary Fig. 11). We believe that these points suggest that condensation might be a useful phenotypic marker of cell death. To make these points clearer to all readers, we have added additional explanations beginning on line 184 of the main text, as well as in the Discussion, beginning on page 18.

We agree with the reviewer that kanamycin and ciprofloxacin are not distinguished by lytic modes of action, but also wish to point out that kanamycin and ciprofloxacin still induce lysis in cells, as we have shown in the present work. Importantly, as mentioned in response (2) above, traditional microbiological time-kill assays suggest substantial killing of kanamycin and ciprofloxacin-treated cells, but this does not imply that a majority of cells has been killed within the ~3 h observation time chosen: rather, these assays imply that, after ~3 h of antibiotic treatment, cells do not grow colonies after plating and overnight incubation. We have shown that, at the single-cell level, cells can continue to accumulate biomass for several hours after antibiotic treatment (Fig. 3 and Supplementary Movies 1-3); recent work by the Zhao and Drlica labs has also shown that self-driving ROS accumulation occurs and induces cell death after plating, in contrast to the idea that cells are dead at the time of plating. As we have now better emphasized beginning on lines 184 and 394, understanding at what point antibiotic-treated cells die was one of the motivations of the present work, which reveals cytoplasmic condensation as a final cell death phenotype.

- 5. Lines 65 and 72: If condensation occurs within one minute, how long does the phenotype persist before lysis occurs? Rapid loss of condensation may explain why only a small fraction of cells could be captured to show the condensation phenotype.**

We have clarified that condensation typically persists for minutes to hours in the main text, beginning on line 84. We agree with the reviewer that cells may transition quickly from condensation to lysis, possibly within seconds or fractions of seconds. In the Methods section, under *Image Analysis*, we have now clarified that because of this, our estimates for the frequency of cytoplasmic condensation is a lower bound which is limited by both the time resolution of our microscopy as well as the total duration of observation.

- 6. Lines 97-102: Why did the increase in SYTOX Blue fluorescence occur at different stages with kanamycin- and ciprofloxacin-treated cells? Does condensation progress into lysis much quicker with kanamycin-treated cells than with ciprofloxacin-treated cells?**

We thank the reviewer for this intriguing question. To address it, we have quantified the times between condensation and lysis in cells and found that kanamycin-treated cells lyse earlier in time than ciprofloxacin-treated cells, which typically remain condensed for longer durations (Supplementary Fig. 9). This suggests that dyes may have more time to intercalate in ciprofloxacin-treated cells than kanamycin-treated ones, consistent with our SYTOX Blue measurements that show increased fluorescence during condensation in ciprofloxacin-treated cells and lysis in kanamycin-treated cells. We also wish to point out that aminoglycosides are known to damage membrane through ionic interactions, which might also contribute to rapid lysis and decreased stability of the condensed phenotype. We anticipate future work using functionally inactivated aminoglycosides, as previously done in ref. 24, to better understand the extent of membrane damage occurring through ionic interactions, as opposed to potential reactive metabolic byproduct-driven processes.

- 7. Lines 135-137 and 142-145: Since cells showing condensation failed to elongate or even filament and since filamentous cells, but not condensed cells, could resume growth after antibiotic removal, condensation must have occurred after the irreversible death step is complete. If so, what is the relevance of condensation to the death process?**

We agree that condensation likely occurs after the irreversible death step, which our results suggest to be associated with membrane damage. Additional experiments with hypoosmotic shocks, as summarized above in response #1, further support that it is membrane damage which lies upstream of cytoplasmic condensation that contributes to cell death. Thus, we believe that condensation is a final manifestation of cell death induced by membrane damage—which in turn occurs downstream of lipid peroxidation and generation of reactive metabolic byproducts—and have made the connection between these processes clearer beginning on lines 184, 362, and 394. We thank the reviewer for bringing our attention to this important point.

- 8. Lines 164-165: How could high ROS fluorescence signals be observed with lysis and condensed cells? The membrane integrity of these cells had already been lost, ROS production should have been stopped, and the ROS-sensitive dyes would have readily leaked out of the broken cells.**

We thank the reviewer for this intriguing question. In our microscopy measurements, we found that many lysed cells still contained pockets exhibiting high fluorescence, ostensibly because of some residual cell material in these pockets. Even when averaged over the 2D cellular area, the fluorescence intensities of these pockets were larger than corresponding fluorescence intensities in turgid cells. An example of this is shown in Fig. 4b.

In the case of condensed cells, we wish to point out that, as suggested by our biophysical model, only nanometer-scaled membrane defects are needed to explain condensation, and that these defects may be large enough to transport solutes but not necessarily larger macromolecules, such as proteins. As reactive metabolic byproducts including ROS and RNS can interact with, and be produced by processes involving, proteins and larger macromolecules, we believe that their detection within the cell is not inconsistent with the condensed phenotype, as underscored by the fluorescence microscopy images shown in Fig. 4a,c and Fig. 7a of the revised paper.

- 9. Lines 187-190: Does glutathione affect MIC of kanamycin or ciprofloxacin? If it does, killing should be done using concentrations normalized to a certain fold of MIC rather than using the same absolute antibiotic concentration. This is important for obtaining a valid comparison.**

We have performed additional MIC experiments in 96-well plates to determine MICs in the presence of 10 mM glutathione. Consistent with prior literature, we found that several MICs were indeed increased; Table R1 lists previously found MIC values and their counterparts in the presence of 10 mM glutathione.

Antibiotic	MIC range in LB only ($\mu\text{g/mL}$)	MIC with LB+10 mM glutathione ($\mu\text{g/mL}$)
Kanamycin	[5.0,12.0]	>50

Ciprofloxacin	[0.03,0.1]	0.25
Gentamicin	[3.13,6.25]	>10
Norfloxacin	[0.15,0.5]	2.5
Ampicillin	[3.13,12.5]	6.25
Mecillinam	[0.63,2.0]	1.0

Table R1: MICs of antibiotics used in this study, with and without 10 mM glutathione. MICs were determined as described in Methods, and ranges are shown from three replicates for each overnight culture dilution (1:100 and 1:10,000) and each growth vessel tested (14 mL Falcon tubes and 96-well plates). MICs in the presence of glutathione were determined for a 1:10,000 overnight culture dilution in 96-well plates, and are representative of two biological replicates.

However, we respectfully wish to point out that antibiotic concentrations used in combination treatments with exogenous compounds have been referenced with respect to MICs without these compounds present; see, for instance, refs. 3, 10, 11, 13, 15, and 24. This is because an MIC increase in the presence of exogenous compounds, as in the case of glutathione, can arise from the exogenous compound making the inoculum in growth inhibition experiments less susceptible to antibiotic by reducing killing. Normalizing to the MIC with the exogenous compound present would control out this treatment effect, which is certainly relevant to time-kill assays.

10. Line 209: The word “many” is a value judgement that needs to be quantitatively supported. What is the percentage of dead cells that exhibit the condensation phenotype?

We have decided to remove “many”, as we feel that the intention of this sentence was to summarize a plausible pathway of antibiotic-induced cell death and not to make a quantitative statement about the frequency of the phenotype observed. For the latter, we have better clarified on line 72 of the main text that ~30% of cells treated with kanamycin and ~90% of cells treated with ciprofloxacin exhibit condensation. We have further noted the limitations of our counting method in response #5 above and in the Methods section of the main text, and better clarified that condensation is not observed in all susceptible cells starting on line 209.

11. Legends of many figures are difficult for general readers to follow. The figures themselves should be easily understood without reference to the Methods or Results sections.

We thank the reviewer for pointing this out and have significantly edited all figure legends to remove extraneous details, add additional clarifications as necessary, and simplify the presentation. We very much hope that this will make our work easier for general readers to follow and thank the reviewer again for their important and thoughtful comments.

Reviewer #2.

We thank the reviewer for their thoughtful and detailed report.

12. In this article, Wong et al investigate the impact of aminoglycosides and fluoroquinolones on E. coli at the single cell level. They report that these two classes of antibiotics indirectly damage the membrane via toxic by-products, which causes the outflow of the cytoplasmic content and cytoplasm condensation, which coincides with loss of cell viability. The authors also report that pre-treating the cells with the antioxidant GSH attenuates cell death. The main message of the paper will be of general interest to the microbiology community and to people active in antibiotic research, in part by suggesting lipid peroxidation as a valid target for antibiotic development. Although I find the work interesting, I have a number of concerns that need to be addressed; the paper currently lacks key control experiments and crucial data.

We thank the reviewer for their insightful comments, which have helped to strengthen the piece considerably, and hope that the reviewer finds their concerns adequately addressed with our introduced revisions and responses included below.

- 13. Cytoplasmic condensation is defined by the authors as regions growing denser; in Fig S2, they used as read-out the density of phase contrast measurements on 18 cells. Another method should be used to better assess cytosolic packing, for instance using a cytoplasmic fluorescent probe like mCherry.**

We have performed additional microscopy experiments with an *E. coli* strain expressing cytoplasmic mCherry, as summarized in Supplementary Fig. 2. Quantitative measurements of intracellular fluorescence in this strain support the conclusion of cytosolic packing during the cytoplasmic condensation we observed. Furthermore, we have augmented all phase contrast and fluorescence microscopy measurements with additional automated measurements using the plugin MicrobeJ, so that there are at least 20 data points for each condition and each experiment in all figures (please also see response (14) below). Finally, to improve reproducibility, we have provided all source data in a separate .xlsx file.

- 14. The number of cells analyzed and the methodology used (manual detection) are far from being optimal. I do not agree with the authors when they justify (lines 438-446) that “pre-existing software, such as CurvatureTracker or MicrobeTracker, did not provide adequate tools for automating the collection of data required”. These tools are in fact versatile and able to detect very complex shapes. If the authors experienced difficulties with these software, they should try a tool like microbeJ which is more user-friendly. Using these tools is important to allow an unbiased and statistically-relevant analysis of microscopic images. Manual detection of the cell edges can lead to approximation, especially if the measurements have been performed as in fig.1e by simply “drawing a straight line” between the two poles without taking into account the bending of the cell and its movement in response to the modification of the composition and flow of the liquid in the chamber of the microfluidics device.**

We thank the reviewer for their attention to this point. We wish to clarify that previous measurements of cell length were performed by drawing a curve which tracks the cellular midline and takes into account cell bending; as only a guide to readers, we had intended for the annotations in Fig. 1e to emphasize how the positions of the poles of different cells react to changes in external osmolarity. We also agree with the reviewer that it is important to have unbiased measurements from the image analysis. To further address the reviewer’s point, we have updated the methodology to use MicrobeJ, as detailed in the Methods section in the main text. Re-doing the image analyses across all figure panels, we found that the results obtained from delineating cell contours using MicrobeJ were quantitatively consistent with our manual results. Thus, where applicable, we have updated all corresponding measurements across figures to reflect the semi-automated pipeline using MicrobeJ. As mentioned in response #13 above, we have also increased the numbers of cells used for all analyses, and we have now provided all data from our measurements as in a separate .xlsx file to aid reproducibility.

- 15. Line 65: the sentence “discrete portions of the cytoplasm typically became phase-light” should be rephrased: these phase-light zones correspond to an increased periplasmic space.**

We have rephrased the sentence to read: “discrete portions of cells typically became phase-light” (line 74); furthermore, the connection between the phase-light region and an increased periplasmic space, to which the reviewer refers, is now better emphasized on line 80.

- 16. Line 70-72: It is unclear if the cytoplasm separates into different sub-compartments which are not connected to each other or not. Could the authors comment on that? Also, did the authors test the reversibility of the plasmolysis they observe?**

We agree with the reviewer and believe that follow-up experiments examining mobility within the subcompartments that form would be interesting. However, in the present work, we were more focused on understanding the implications of condensation on cell death. We felt that Supplementary Note 1, in which we

discuss the model and its predictions relevant to cellular physiology, was the appropriate place to comment on this and have appended an explanation as the last paragraph of Section 1.2.4 there.

We also note that our microfluidic experiments, in which antibiotics were washed out after application, indicated that condensation could not be reversed by the absence of antibiotics alone. To further address the reviewer's question, we performed additional microfluidic experiments involving hypoosmotic shocks. In these experiments, cells were grown in hyperosmotic media comprising LB and 250 mM sorbitol while treated with kanamycin or ciprofloxacin (10x MIC). At the onset of cytoplasmic condensation, cells were hypoosmotically shocked by flow of LB+drug only. We found that a small fraction (~10%) of condensed, ciprofloxacin-treated cells briefly became turgid, as indicated by phase-light regions becoming phase-dark and increases in cell size, so that hypoosmotic shock transiently reversed cytoplasmic condensation in these cells. However, after several minutes—a timescale consistent with the solute outflow predicted by our model—all previously condensed cells again manifested the condensed phenotype, and all such cells failed to elongate upon continued observation. In contrast, kanamycin-treated cells rapidly lysed after hypoosmotic shock, suggestive of more widespread membrane damage. Taken together, these observations suggest that physical changes in osmolarity do not modulate kanamycin and ciprofloxacin-induced cell death and membrane damage. We have added a discussion of these new experiments starting on line 362 of the revised paper.

17. Lines81-95: turgor pressure is caused by an osmotic flow of water. A loss of turgor pressure induces an exit of water from a compartment of lower solute concentration into a compartment of higher solute concentration. Is the extra-cytoplasmic environment of the condensed cells hyper osmotic?

We agree with the reviewer's interpretation and wish to point out that our biophysical model of solute outflow (1) quantitatively captures the loss of turgor and (2) indicates that solutes are transported from the intracellular to the extracellular environments. Our osmotic shock experiments (Fig. 1e), in which condensed cells that were hyperosmotically shocked displayed no contraction, support the notion that the turgor pressure is smaller in these cells than in turgid cells.

To further support our model prediction of solute outflow, and related to point (18) below, we have performed additional microscopy measurements with application of the potassium ion-sensitive dye ION Potassium Green (IPG). IPG is a membrane permeable dye that fluoresces upon binding potassium ions, which comprise one of many species of intracellular solutes. Using IPG, we found that condensed cells generally exhibited little fluorescence compared to turgid cells, indicating that condensed cells contain lower concentrations of intracellular potassium. These additional measurements have been added to Supplementary Fig. 6 and discussed beginning on line 98 of the revised paper.

18. The authors propose that reactive by-products cause membrane damage, then cytoplasmic solutes go out, and finally the cytoplasm becomes condensed. But, what is the evidence that solutes go out? This should be tested using small cytoplasmic fluorescent proteins for instance. Further experiments are required to address this and clarify the mechanism.

We believe that the combination of (1) osmotic shock experiments indicating that the turgor pressure of condensed cells are smaller than that of turgid cells; and (2) additional measurements measuring potassium ion concentration with IPG, as detailed above in response #17, provides evidence toward addressing the reviewer's point. We also wish to point out that our biophysical model suggests that the membrane defects generated before cytoplasmic condensation can be small—with nanometer-sized radii—and need not be large enough to accommodate the outflow of cytoplasmic proteins, which nevertheless would be interesting to test in future studies.

19. Why do stationary-phase cells show less cytoplasmic condensation (lines149-151)? Were non-condensed treated cells in exponential phase not elongating? Does a slower growth rate negatively impact cytoplasmic condensation? This needs to be discussed.

We have added additional explanations to the main text, that (1) stationary phase cells are metabolically dormant and less susceptible to antibiotics (beginning on line 214); and (2) non-condensed, treated cells in exponential phase were elongating, as shown in Fig. 3b. As suggested by point (1), we believe that a slower metabolic rate negatively impacts cytoplasmic condensation by decreasing the susceptibility of cells to antibiotics, as previous work from both our labs and others have shown (please see, e.g. refs. 24 and 25).

20. Membrane damage is crucial to explain the cause of death. Therefore, the authors need to test the validity of the various dyes they used as a proxy for membrane damage using normal plasmolysed cells: these cells should have no membrane damage.

We thank the reviewer for pointing this out and have added additional control measurements of these dyes for untreated cells that are plasmolyzed (but not lysed) through application of a 500 mM hyperosmotic shock. As expected, these cells did not exhibit membrane depolarization or damage (Supplementary Fig. 8).

21. Clearly establishing lipid peroxidation is also very important to validate the mechanism. The observation made using the fluorescence probe is indirect. A lipidomic study should be done to characterize the membrane damage that occurs and causes the leak. Also, the authors should test the impact of carrying out the experiments anaerobically to see what happens in the absence of oxygen. Did the authors try adding fatty acid to rescue the condensed cells?

We thank the reviewer for this insightful suggestion and have performed several additional experiments to address it. First, we have assayed changes in lipid composition in antibiotic-treated cells. Consistent with our observations of membrane damage suggested by SYTOX Blue, we found that free fatty acid levels were increased in all antibiotic-treated cells (Fig. 5a,b). Intriguingly, we also observed salient shifts in cardiolipin (CL)/phosphatidylethanolamine (PE) ratios (Fig. 5a,c), which have previously been shown to be associated with changes in cell envelope ultrastructure similar to that found for cytoplasmic condensation here (ref. 30).

Next, we performed additional experiments in an anaerobic chamber. Here, cells were grown, treated, plated, and incubated in LB under strictly anaerobic conditions. We found, intriguingly, that the absence of oxygen alone does not significantly alter antibiotic killing or the occurrence of cytoplasmic condensation (Fig. 7a,b). As suggested by previous work from our lab (ref. 13), we believe that this could be caused by the presence of other reactive metabolic byproducts, including but not necessarily limited to reactive nitrogen species, which may contribute to lipid peroxidation and membrane damage. Consistent with this hypothesis, anaerobically imaging cells in the presence of C11-BODIPY revealed significant fluorescence in ciprofloxacin- and kanamycin-treated cells, but not untreated cells (Fig. 7a).

Finally, we attempted to rescue antibiotic-treated cells by exogenous supplementation of the lipophilic antioxidant α -tocopherol (50 mM), which has been evidenced to mitigate lipid peroxidation in eukaryotes. However, we found that this did not alter antibiotic lethality (Fig. 6g). Consistent with our finding that cytoplasmic condensation is associated with cell death, supplementation of α -tocopherol also did not alter the emergence of cytoplasmic condensation in antibiotic-treated cells (Fig. 6h). We speculate that this method of mitigating lipid peroxidation may be less potent in *E. coli* than in eukaryotes. Consistent with this hypothesis, microscopy measurements showed that C11-BODIPY still exhibited substantial fluorescence in cells treated by exogenous α -tocopherol (Supplementary Fig. 15). Intriguingly, however, we note that recent work (ref. 38) applying an α -tocopherol analogue has reported decreased ciprofloxacin killing and alleviated lipid peroxidation in *E. coli*. We have pointed out this work and summarized our additional findings beginning on line 327 of the main text of the revised paper.

22. Line173: A positive control is missing: what is the impact of adding an external source of peroxyinitrite to the untreated cells? Does combining peroxyinitrite and either Kan or CIP further increase the number of condensed/lysed cells?

We thank the reviewer for this comment and have performed additional time-kill and microscopy experiments to address it. We found that exogenous supplementation of peroxyinitrite (1 mM) decreased cellular viability

by ~1 log after 2 h of treatment (Fig. 6e). Intriguingly, peroxyntirite treatment also induced cytoplasmic condensation, and combining peroxyntirite and kanamycin or ciprofloxacin treatment subadditively increased the number of condensed cells, by ~10-20% (Fig. R1).

Fig. R1: Frequency of cytoplasmic condensation in two fields of view in cells treated by antibiotics (10x MIC) and/or peroxyntirite (1 mM) for 6 h. Frequencies from individual fields of view are indicated by points, and were determined as detailed in the Methods. Two-sample *t*-tests for differences in mean value between antibiotic+peroxyntirite and antibiotic or peroxyntirite indicated $p < 0.05$ for all four tests. Untreated control cells exhibited no condensation or lysis (see, e.g., Fig. 1d of the main text).

Together, these findings are consistent with the hypothesis that reactive metabolic byproducts, including but not limited to peroxyntirite, can contribute to the condensation phenotype.

- 23. The authors should test whether *gor* or *gshA* mutants are more sensitive to antibiotic treatment. Also, did they try using other antioxidants, like DTT or mercaptoethanol? Is the lethality of the peroxyntirite-treated cells rescued by the addition of glutathione? Also, in Fig 4i (with KAN), representative images do not suggest that “At the single-cell level, glutathione largely alleviated condensation and lysis across a range of treatment conditions”**

We have performed additional time-kill experiments and microscopy to address this comment. Cells with *gor* and *gshA* deletions from the Keio collection were not substantially more susceptible to antibiotic killing, as evidenced by CFU quantitation after 4 h of treatment (Supplementary Fig. 14); we speculate that these genetic perturbations induce smaller variations in glutathione concentration than are needed to protect against antibiotic lethality, as endogenous pools of glutathione have been estimated to be ~10 mM in *E. coli* (ref. 35). Intriguingly as well, cells pretreated with DTT and mercaptoethanol (10 mM) were rescued from peroxyntirite-induced killing and cytoplasmic condensation, but not from antibiotic-induced killing and cytoplasmic condensation. We believe that this difference, not observed in glutathione pretreated-cells that are tolerant to both peroxyntirite and antibiotics, could arise from different scavenging potentials, as suggested beginning on line 313 of the main text of the revised paper. We anticipate a future study in which the effects of different antioxidants are more systematically explored.

We thank the reviewer as well for pointing out the phrase mentioning Fig. 4i, and have rephrased it to read: “At the single-cell level, glutathione pretreatment was accompanied by cell proliferation in kanamycin-treated cells and notable suppression of condensation and lysis in ciprofloxacin-treated cells (Fig. 6c,h and Supplementary Movies 6 and 7), suggesting a reactive metabolic byproduct-based origin for the membrane damage and killing observed in the absence of glutathione” (line 291).

- 24. The mode of death of the antibiotic-treated cells is reminiscent of that of cells with a dominant mutation in MlaA (MlaA*; Sutterlin et al, 2016, PNAS). In the 2016 study, the authors argue that these phenotypes result from an increase of LPS in the outer membrane and from a futile flow of lipids from the inner to the outer membrane. Did the authors consider a loss of lipids from the inner membrane and did they check the LPS levels?**

We thank the reviewer for pointing out this interesting work by labs that we very much appreciate. There, the authors propose, in addition to leakage of cytoplasmic contents, that membrane damage is induced downstream of an increase of LPS in the outer membrane. We had previously not considered loss of lipids from the inner membrane but have performed additional experiments and analyses to (1) assay the LPS levels of bulk cultures using an LAL assay; and (2) check for the presence of membrane blebs, which the authors of the aforementioned work proposed to manifest increases in LPS levels, in microscopy images. We observed no substantial membrane blebs in all our microscopy experiments; representative fields of views in phase-contrast are presented in Supplementary Fig. 4 and the Supplementary Movies. Imaging with fluorescence microscopy with all the cell envelope-marked strains specified in the paper also did not reveal the presence of substantial membrane blebs.

Using an LAL assay at the bulk culture level, we found intriguingly, that kanamycin- and ciprofloxacin-treated cells exhibited lower LPS concentrations than untreated cells (Supplementary Fig. 13). Intriguingly, this suggests that increases in LPS levels may not be associated with cell death in cells treated with kanamycin and ciprofloxacin. We have added a discussion of this finding starting on line 272, and we anticipate further studies to better characterize the differences between the cytoplasmic condensation we observe here and the cell death phenotype characterized in the elegant work pointed out by the reviewer.

- 25. The author should discuss the death mechanism of the fraction of the population that does not display cytoplasmic condensation.**

We appreciate this important point raised by the reviewer and have added additional text in the Discussion to discuss possibilities for the death mechanism of the fraction of the population in which we did not observe cytoplasmic condensation. We note that many cells we observed proceeded directly to lysis or simply did not elongate, as may be consistent with reactive metabolic byproduct-mediated mechanisms that react with nucleic acids and proteins in addition to membrane lipids.

- 26. Fig. S3: the quality of the fluorescence signal is poor and not interpretable.**

We thank the reviewer for pointing this out and note that this particular image was taken on the Zeiss AxioScope A1, which often produces poorer signals when measuring fluorescence in cells with lower copy number fluorophores (as in the case of fluorescently tagged membrane proteins) due to the camera's limited sensitivity. We have (1) replaced the representative image shown with an image from the Nikon Ti inverted microscope, which has a more sensitive camera and TIRF fluorescence; and (2) redone all fluorescence microscopy experiments involving cells with low copy number fluorophores exclusively on the Nikon Ti inverted.

Reviewer #3.

We thank the reviewer for their enthusiastic and insightful report.

- 27. The manuscript by Félix Wong et al. describes the study of how antibiotics induce cytoplasmic condensation through membrane damage and subsequent outflow of cytoplasmic contents. Topic and relevance of results match Nature Communications very well. The manuscript is well written although highly condensed at some places more in the Nature style. For Nat. Communications, as the space limit**

is not so strict, I would strongly suggest to the authors to explain more in details the experiments and the results to facilitate its reading.

We thank the reviewer for their interest and thoughtful suggestions, and have rewritten the main text to better explain our experiments and our results. We very much hope that the revised manuscript is now more accessible to readers.

28. Lines49-61: Why is the duration of treatment different between AMP (1h), CIP (6h) and KAN (3h)?

We apologize for not making this clear. We have further emphasized in the revised paper that the phenotypic changes induced by these three different antibiotics are different and occur on different timescales—albeit cells being treated uniformly at 10x the MICs—ostensibly due to their different primary binding targets: ampicillin induces rapid membrane bulging and lysis within ~1 h, while kanamycin and ciprofloxacin induce cytoplasmic condensation and lysis on longer timescales of several hours. To better accentuate the differences in observation time, we have better explained that kanamycin- and ciprofloxacin-treated cells do not exhibit substantial phenotypic change before condensation, other than filamentation, in the caption of Fig. 1a and beginning on line 57 of the revised paper.

29. When looking at Fig 1a and in particular to the "untreated cells" (see Zoom#1; attached), it seems obvious that some of the untreated cells are also lysed (cells a much less dark & with some darker parts localized at the poles or in the septum area). This means that these effects cannot only be exclusively attributed to the antibiotics itself. Surprisingly in Fig 1d, for the untreated cells the population fraction of lysed cells remains at "0". How do you explain this? As a control the untreated cells should also be labelled with the fluorescent markers as in Fig 1b.

We thank the reviewer for pointing this out. We had indeed considered the untreated cells shown below the "+KAN" panel as phase dark, but as the reviewer points out, certain cells exhibited local variations in phase contrast. We believe that this variation is expected in phase-contrast images of exponentially growing cells, especially when cells are crowded against each other and may lift off against the glass slide, as we believe to be the case at present. It is also possible that inexact alignment of the microscope condenser (on the Nikon Ti, which was used to take the image) contributed to accentuating these local variations, since they were less noticeable when imaging with a different microscopy setup (on the Axioscope A1).

Nevertheless, to confirm that these cells are not lysed, we verified that staining with SYTOX Blue does not produce any fluorescence signal in untreated cells exhibiting similar variations in phase contrast (Fig. 1g). We have further updated the microscopy images shown throughout the work to show fields of view without large crowds of cells. Lastly, we have added better representative microscopy images to Supplementary Fig. 3.

30. Fig 1d. Is this quantification performed based on the phase contrast image? As shown in Fig 1b, for some cells (last row) the condensation on the phase contrast image is not evident, but on the contrary much easier to observe on the fluorescence image. Therefore this quantification would be more accurate if done on the fluorescence images.

We thank the reviewer for this suggestion and agree. We have now better distinguished the condensed and lysed phenotypes using both phase contrast in the MG1655 strain and phase contrast and fluorescence in the cytoplasmic mCherry strain, as detailed beginning on line 601 in the Methods. We found that the frequency of condensation and lysis events were similar in these two strains across time and have pooled all the data in the revised figure. To improve reproducibility, we have also provided all source data in an accompanying .xlsx file.

31. Fig1f: The authors analyzed by AFM the elastic modulus of cells treated or not with CIP. What is the applied force used to probe the elastic modulus? Which model did they used? Could the authors provide in SI some force-indentation curves with the fitted model? How was calibrated the spring constant of the tip? As the authors used the QI mode, they should also have access to the simultaneously

recorded elasticity map showing the distribution of the Young modulus. This should also be presented to further evidence the localization of the different part of the cell with altered elastic properties. Are the results similar for the cell treated with KAN?

We have better described and provided additional force-indentation curves, details regarding calibration, experimental setup, and model, and elastic modulus heatmaps in the Methods and Supplementary Fig. 7 of the revised paper. In particular, the applied force used is 2 nN, the model used is the Hertz model for a paraboloid indenter, and the cantilever was calibrated using routines within the AFM software based on the Sader method (ref. 49).

We have also performed corresponding measurements for cells treated with kanamycin (10x MIC). Intriguingly, we observed similar, but less pronounced, variations in cell stiffness in kanamycin-treated cells, whose cell-averaged elastic moduli are not significantly changed relative to untreated cells (Supplementary Fig. 7). This finding contrasts with our observations for ciprofloxacin and suggests that cells may exhibit both increased (ciprofloxacin) or roughly similar elastic moduli (kanamycin) after condensation, which is associated with the loss of cellular turgor. It is possible that this difference between condensed ciprofloxacin- and kanamycin-treated cells could arise from additional membrane damage through ribosome-independent ionic interactions of aminoglycosides with the outer membrane, which contributes to destabilizing the cellular envelope (ref. 20). Additionally, another possibility suggested by our fluorescence intensity measurements (Supplementary Fig. 2) is that, while cellular turgor is decreased in condensed cells treated by either antibiotic, less intracellular compactification of macromolecules occurs in kanamycin-treated cells. We anticipate future work to better clarify the source of this difference between kanamycin- and ciprofloxacin-treated cells, for instance by using functionally inactivated aminoglycosides, as previously done in ref. 24, and we have revised our discussion of the AFM experiments starting on line 110 in the main text to carefully distinguish these differences.

32. The size of the cell before and after treatment with antibiotics was studied by light microscopy. However, the resolution of such a method is quite low. As the authors have access to an AFM, it would be interesting to analyze the length and height (diameter) of cells (untreated or treated with CIP and KAN) by AFM. It would be interesting to see whether or not the decrease in length is associated with a change in diameter.

We have performed additional experiments and analyses to address this thoughtful suggestion. We used AFM to analyze the length, height, and diameter of untreated cells and antibiotic-treated cells (10x MIC), as shown in Supplementary Fig. 7e. We found that, consistent with phase contrast microscopy observations, the overall dimensions (height, length, and diameter) of ciprofloxacin-treated cells were increased compared to untreated cells, while the dimensions of kanamycin-treated cells, aside from diameter, were not significantly different from untreated cells. However, we would like to point out that nearly all ciprofloxacin-treated cells (and some kanamycin-treated cells) exhibit filamentation, as shown in Fig. 1a of the main text; in light of this, we believe that the observed increases in cellular dimensions (especially length, but also width and height) in antibiotic-treated cells, relative to untreated cells, may be confounded by filamentation.

As alluded to in response #35 below, we believe that another experiment is to measure the dimensions of the same, antibiotic-treated cell immediately before, and after, cytoplasmic condensation. However, we have found this experiment difficult to perform on our AFM setup, as it involves treating cells with antibiotic after washing, during AFM, and requires extended periods of observation in order to find cells that condense during the timeframe of the experiment. We have therefore performed additional fluorescence microscopy experiments, in which the cytoplasm of cells were fluorescently tagged and used to determine cell width during condensation. We note that timelapses of condensed cells from these experiments are in good quantitative agreement with phase-contrast timelapses, and both indicate that the cell length and cell width decrease during condensation, consistent with our model and the overall decrease of cell volume inferred. We have provided additional traces for cell length, width, and volume in Supplementary Fig. 9b and hope that this sufficiently addresses the reviewer's comments.

- 33. Line 89-98. From the hyperosmotic shocks experiments, the authors conclude that the turgor of condensed and lysed cells is significantly diminished compared to turgid cells. However the AFM experiments suggest the opposite. As an increase in the elastic modulus is observed, it means that the cells are more rigid suggesting an increase in the turgor pressure. This should be clarified. The authors could use biophysical model to extract the turgor pressure from AFM indentation experiments performed on the bacteria cells?**

We thank the reviewer for pointing this out and have clarified this point in the main text, beginning on line 110. Indeed, our AFM observations suggest that condensed cells are stiffer than untreated controls, and it is possible that decreases in effective stiffness, arising from the loss of turgor pressure associated with cytoplasmic condensation, may be offset by the intracellular compactification of macromolecules suggested by phase-contrast and fluorescence microscopy measurements (Supplementary Fig. 2). This possibility is consistent with *E. coli*'s estimated turgor pressure of ~1 atm, which is small compared to the magnitude of stiffness increase reported in condensed, ciprofloxacin-treated cells (~5 atm; please see Fig. 1f and Supplementary Fig. 7).

We agree with the reviewer that it would be interesting to extract the turgor pressure from AFM experiments. We are only confident doing so for untreated cells, due to the confounding factors of turgor pressure loss and intracellular compactification reported here in antibiotic-treated, condensed cells. Adapting Boulbitch et al.'s seminal work (ref. 50) to our experiments, we found that, in untreated cells, the inferred turgor pressure is ~1 atm (~100 kPa). This estimate is qualitatively consistent with recent AFM experiments by the Shaevitz group (ref. 51) and approximates turgor pressure estimates from other osmolarity-based studies well (ref. 19); details of the calculation have been added to the Methods (line 549) and mentioned in the main text, on line 116.

- 34. The biophysical model is very interesting. What is the evidences of the nanometer scale membrane defects? An alternative model could be that the condensation of the cytoplasm could lead to an increase in the water activity and therefore water outflow leading to a decrease in the cell volume. By comparison, this is something observed for mammalian cells just before mitosis, during pre-mitotic condensation. After the chromatin condensation, the cell volume decreases. This process involves the outward directed movement of chloride followed by water.**

We thank the reviewer for their interest and have better explained the rationale for the nanometer-scale membrane defects in the main text beginning on line 132, in particular noting that (1) SYTOX Blue only penetrates cells with compromised membranes and (2) the resulting loss of turgor induced by membrane damage is consistent with our osmotic shock experiments. We have also performed additional experiments with IPG, a membrane-permeable potassium-sensitive fluorescent dye, to support the hypothesis of solute loss in condensed cells (Supplementary Fig. 6).

We agree that water outflow is important in these processes and think it is very likely the case that both water and solute transport is needed to explain our observations, as detailed in Supplementary Note 1. We are very much intrigued by the reviewer's insightful reference to mammalian cells. We would respectfully add that a major difference between bacteria and mammalian cells is that the latter typical does not sustain a large turgor pressure. This is important because we believe that the outflow of water from a bacterium, by itself, would only increase the bacterium's turgor pressure (due to the entropic origin of turgor; please see Supplementary Note 1). As such, this prediction, by itself, seems inconsistent with our osmotic shock experiments (Fig. 1e and Supplementary Fig. 5) and our IPG experiments (Supplementary Fig. 6).

- 35. In figure 2f, the model is compared to the empirically observed decreases in cellular volume. How do you measure the volume experimentally? Since you observed a 20% decrease in length, do you assume that the radius remains constant? This must be validated by AFM experiments to have precise estimate of the radius before and after treatment. It would also be useful to validate the biophysical model.**

We thank the reviewer for pointing this out and have better explained the volume measurements in the Methods (line 601); in particular, we assume that the radius remains constant along the cellular midline, but

not necessarily in time (before and after condensation). We have now shown cell width data before and after condensation, as measured by image analysis on both phase contrast and fluorescence microscopy timelapses (Supplementary Fig. 9 and Methods). We hope that this addresses the reviewer's comment and thank the reviewer again for their interest and insight.

Reviewer #4.

We thank the reviewer for their positive and thoughtful report.

- 36. I like the manuscript in general, especially the variety of targeted single cell experiments combined with rigorous biophysical modeling. To my knowledge, the major point of the paper addressing drug induced cytoplasmic condensation is novel. The Methods are detailed and well written. I would thus recommend publication after the following concerns/questions have been addressed:**

We are glad that the reviewer appreciates the manuscript and hope that their comments have been adequately addressed by the introduced revisions and additions.

- 37. The authors argue that metabolic byproducts (free oxygen radicals) may cause lipid peroxidation that results in rupture of the membrane. Their experiments seem to support this hypothesis, however, this still leaves the question how a hole forms in the cell wall. Or can the cell wall be considered porous enough for the cytoplasm to flow through basically unhindered?**

We wish to point out that the cell wall, a peptidoglycan mesh, is inherently porous: elegant AFM studies by Simon Foster's group have characterized the architecture of the cell wall, which exhibits characteristic pore sizes with areas of $\sim 10 \text{ nm}^2$ (ref. 21). As these pore sizes are larger, or of a similar scale to, the membrane defects we consider in our work, we anticipate that the cell wall is not limiting for the flow of solutes we model. We have better explained this assumption, as well as other assumptions of our model, in the main text beginning on line 153 and thank the reviewer for pointing this out.

- 38. Similarly, the mathematical model assumes that straight holes are formed that extend through all three layers of the cell envelope. I am guessing this is just a simplification for modeling purposes as Fig. 1c depicts holes in the inner and outer membrane that do not overlap? Independent of that, the picture seems to be that static holes are being created in the cell envelope. I would argue that membrane fluidity rapidly closes these holes ($\sim 1 \text{ sec}$). I think cells undergoing electroporation or chemical transformation face a similar situation: small holes are being created such that DNA can pass through, however, the cytoplasm does not condense, and most cells survive. Is the idea behind the model that there are many holes that open and close constantly? I don't think this is reflected in the mathematical model. Could the authors please elaborate?**

We thank the reviewer for this intriguing and important question. Our model does not account for dynamics in membrane permeability and does not necessarily make an assumption to whether the holes close or not; rather, the model assumes a constant number of holes for simplicity, and that they could, in principle, be anywhere on the cell envelope. We believe that the combination of our SYTOX Blue measurements and our osmotic shock experiments, respectively, suggest that the membrane, at some point in time, is indeed comprised; and that the membrane is comprised to the extent that significant outflow can occur. To verify this latter point, we have performed additional microscopy measurements with application of the potassium ion-sensitive dye ION Potassium Green (IPG). IPG is a membrane permeable dye that fluoresces upon binding potassium ions, which comprise one of many species of intracellular solutes. Using IPG, we found that condensed cells generally exhibited little fluorescence compared to turgid cells, indicating that condensed cells contain lower concentrations of intracellular potassium. These additional measurements have been added to Supplementary Fig. 6 and discussed beginning on line 98 of the revised paper.

We also appreciate the reviewer's comparison to electroporation and other processes, and believe that our model is not inconsistent with membrane fluidity. Indeed, membrane fluidity allows membrane lipids to reorganize in-plane to assume various shapes, and this is incorporated into the model, in part, by having the membrane be fluid and characterized by a stretching modulus (in contrast to the cell wall, which is modeled as a rigid, elastic shell). Additionally, we believe the main difference between the defects we describe and the small holes generated during electroporation and other similar processes is that, in our case, the membrane area is limited, such that in-plane stretching of the membrane and overall solute leakage still occurs across time (though not necessarily through the same defects); and in transformation experiments, lipid reservoirs and excess membrane area may patch smaller membrane defects and limit solute outflow. In any case, we agree that we have assumed, based on our empirical observations, that the membrane is compromised across time, and we have better described this assumption beginning on line 162 of the main text.

Finally, we enthusiastically agree with the reviewer that it would be interesting to probe whether the same membrane defects stay open or dynamically form and close, but feel that this would be best relegated to future collaborative work in which fluctuations in membrane permeability can be more accurately determined, for instance with patch-clamp experiments. We concur that additional elaboration is needed in light of these simplifying assumptions. In the meantime, we have better explained our model and these assumptions in the main text beginning on line 153 and hope that this sufficiently addresses the reviewer's comment.

39. The mathematical model to describe the temporal outflow of solute seems sound, justified by the excellent agreement with experimental data shown in Fig. 2f and consistency checks shown in the Supplement. I think, however, some of the simplifications used to simplify the math should be rationalized better. Firstly, the authors assume that there is no water in the periplasm, which I think is not entirely correct. The water content of the periplasm also seems to be dependent on the turgor pressure (Sochaki 2011 *Biophysical Journal* - Protein Diffusion in the Periplasm of *E. coli* under Osmotic Stress).

We thank the reviewer for this interesting comment, which has helped bring into focus the assumptions underlying our model. We indeed assume that the periplasm does not hinder the transport of solutes and water out of the cytoplasm. This assumption may be consistent with both (1) a mechanically rigid periplasm; and (2) an isosmotic periplasm, to which the reviewer refers. In the latter case, the cellular turgor pressure drop occurs outside of the inner membrane, and the most salient corrections to the model are that (1) the inner membrane does not become load-bearing; and (2) solute and water transport need only occur across the outer membrane, and not both the inner and outer membranes.

Nevertheless, we wish to point out that multiple lines of evidence support the former assumption, which is the one we have focused on in our model. Previous studies by some of us [S. Hussain *et al.*, *eLife* **7**, e32471 (2018)] and others [S. Wang and N. Wingreen, *Biophys. J.* **104**, 541-552 (2013)] have proposed that the periplasm is effectively a rigid (but permeable) body that mechanically supports the cell membrane. In brief, the rationale underlying this assumption is that the case of an isosmotic periplasm would, at equilibrium, be inconsistent with the existence of a periplasmic space, due to the fact that the bending energy of the inner membrane is minimized when the inner membrane squeezes out the periplasmic space. Additionally, if the periplasm were isosmotic and flow of solutes and water occurred from the periplasm to the extracellular milieu during condensation, then the cytoplasm should be free to expand, due to its large turgor pressure. We find, in contrast, that cells typically shrink during condensation (Fig. 1a, Fig. 2f, and Supplementary Fig. 9). In any case, we agree with the reviewer that a better explanation of our assumptions is needed. We have therefore more carefully listed several salient assumptions underlying the model, including the assumption regarding the periplasm discussed above, in Section 1.2 in Supplementary Note 1, and added additional text describing their rationale.

40. The mathematical model further assumes that all three layers of the cell envelope are disconnected. However, there are many protein complexes such as secretins that span the entire envelope and basically pin them together. Similarly, the model doesn't include the load bearing effect of cytoskeletal elements. Are these negligible?

The reviewer is correct that the three layers are likely to be coupled by membrane-cell wall anchors, cell wall synthases, and other molecules. The question then becomes whether such anchors may affect the energetic calculations considered in our work, either by imposing an energetic barrier of bond breakage or otherwise. We argue, however, that membrane reorganization would simultaneously allow such anchors to be preserved while permitting large-scale variation in the membrane reference states. Provided that the anchors are not sufficiently dense, free phospholipids could slide past the anchors as to remodel the membrane. Indeed, characteristic estimates for the number of free phospholipids in a cell suggest that free phospholipids are much more abundant (by at least ~10-fold) than anchors, and this argument is now discussed at the beginning of Section 1.2 in Supplementary Note 1. We thank the reviewer for drawing our attention to this point.

The reviewer is also correct that our study does not explicitly model the load-bearing effects of cytoskeletal elements, such as the actin homolog MreB. MreB in particular has been shown to be disjoint, but has been evidenced to contribute to cell mechanics, as shown in elegant work by the Shaevitz, Gitai, and Wingreen labs (please see, e.g., ref. 9 in Supplementary Note 1). In light of these pioneering studies, we believe that the mechanical effects of cytoskeletal elements can be accommodated through their contributions to cell wall elasticity. We have therefore viewed these contributions to be coarse-grained, by considering the elastic modulus of the cell wall to be an effective one. We have better explained this assumption at the beginning of Section 1.2 in Supplementary Note 1.

- 41. I have trouble with some of the statistical testing. For example, Fig. 4a,b,c: The results for untreated and turgid are mostly marked as * or ** significance, but the plots shown don't seem to reflect that. Similarly and looking at Supplementary Table 3, it seems that both means of the respective comparisons are identical within much less than one standard deviation. Could the authors please explain? I think it might make sense to show the full distributions in scatter plots, maybe as additional Supplement.**

We thank the reviewer for pointing this out and have revised all bar plots in the main text to show either box-and-whisker plots or individual points, which we hope better illustrate the distribution of data. Indeed, based on the distribution of the underlying data, we wish to point out that statistically significant differences may be distinguished between two different treatment groups even if the respective means lie close to each other, as long as sufficient statistics and small enough variances are observed. In contrast, data points with different means but large variance are less readily distinguished by the statistical tests used. To improve data reproducibility, we have also provided source data in an accompanying .xlsx file. We very much hope that these revisions aid interpretation of our data and the statistical tests used.

- 42. The inclusion of ampicillin seems a bit confusing since it is not actually studied. Is this meant to be a sort of control for an antibiotic that does not induce cytoplasmic condensation?**

We agree that the inclusion of ampicillin is orthogonal to our work, but felt it best to consider ampicillin as a point of contrast to our results for kanamycin and ciprofloxacin. Indeed, our inclusion of ampicillin was motivated in part by the hypothesis that all three antibiotics generate reactive metabolic byproducts (ref. 3 and Supplementary Fig. 1) which may contribute to antibiotic lethality. To better motivate our consideration of ampicillin, we have better explained our rationale for including ampicillin and focusing instead on cytoplasmic condensation induced by aminoglycosides and fluoroquinolones, in the Introduction and on line 68 of the revised paper.

- 43. I like the timelapse plots (Fig. 1i, Fig. 4d) that correlate cytoplasmic condensation with membrane damage, depolymerization, and lipid oxidation. However, the presented intervals of 10 min make it hard to differentiate correlation and causation, especially since the experiments in Fig. 1d and results of the mathematical model suggest that these things happen on the minute or even seconds time scale. I'd love to see at least one plot with a much improved time resolution.**

We are pleased that the reviewer likes the plots and have added additional timelapse images to show condensation and fluorescence of C11-BODIPY dye over a period of a minute, in Fig. 4d. We hope that this sufficiently addresses the reviewer's comment.

44. Although the authors report that the difference in elastic modulus between untreated cells and the phase light region of treated cells is not significant, it still seems that phase light regions are at least as stiff as untreated cells. I would have expected a significant decrease. Can the authors explain?

We thank the reviewer for pointing this out. We believe that the main difference between untreated (turgid) and condensed cells is the leakage of solutes in the latter, which results in (1) a smaller turgor pressure, but (2) intracellular compactification of macromolecules, as suggested by Fig. 1e and Supplementary Fig. 2. We believe that the effects of the latter on stiffness dominate the effects of the former, as suggested by Fig. 1f, in which phase-dark regions of condensed, ciprofloxacin-treated cells exhibit significantly increased elastic moduli. Because there does not appear to be as much cytoplasmic content in phase-light regions, it is unsurprising that the elastic moduli of phase-light regions is substantially less than those of phase-dark regions. However, because typical estimates of bacterial cellular turgor are ~100 kPa, as confirmed here with AFM (Methods), we also do not anticipate a significant difference between the elastic moduli of untreated turgid cells and those of phase-light regions, which have both been measured to be ~600 kPa, especially if some degree of cytoplasmic compactification still occurs in phase-light regions. We have better explained these points regarding AFM measurements starting on line 110 of the main text.

45. In Fig. 1b, the outer membrane label looks more like a cytoplasmic label, especially compared to the inner membrane label that seems to localize much better to the cell envelope. How is this possible?

We thank the reviewer for this comment. We wish to point out that the fluorescence of the outer membrane-labelled strain we used has been documented before in work by the Bernhardt lab (ref. 43). There, the authors showed similar fluorescence across the cellular body, consistent with our observations. One subtle difference to which the reviewer alludes is that the cells in our images do not appear to fluoresce brightly at the cell contours. We have performed additional fluorescence microscopy, using both the Nikon Ti and Zeiss AxioScope A1 setups described in Methods, on this strain to ascertain how this difference might arise. We found, intriguingly, that while the strain was fluorescent on both microscopy setups, the contours fluoresce more brightly when imaged with the lower-powered AxioScope A1, while microscopy on the Nikon Ti reproduced images similar to that shown in Fig. 1b (please see also Fig. R2). We believe that a likely source of this difference is that, for the Nikon Ti scope we used, fluorescence is handled through a TIRF setup. In TIRF, fluorophores that are closer to the imaging interface (the coverslip) fluoresce significantly more strongly than those that are not: in the present case, outer membrane-bound fluorophores that are along the bottom half of the cell fluoresce more strongly than those at the cell contours. This is seen to a similar, but less noticeable, extent in the inner membrane mCherry strain shown in Fig. 1b. In light of this, we have retained the original images in Fig. 1b, as to be consistent with the imaging setup used for all other images in the figure, but have better explained the Nikon Ti TIRF setup in the Methods, as well as the different fluorescence modalities.

Fig. R2: Comparison of images taken on the Nikon Ti and Axioscope A1 for the outer membrane-GFP strain of *E. coli*. Cells were treated by 10x MIC ciprofloxacin for ~2 h before imaging. Scale bar, 3 μm .

- 46. In line 113 the authors introduce their mathematical model without any explanation of what exactly is being model. I think the authors did a great job laying out the model in the Supplement and I encourage them to add a brief summary (2-3 sentences) in the main text.**

We thank the reviewer for their kind words and their appreciation of the model, and have better summarized its aims in the main text on lines 153-161. As detailed in response (38) above, we have also better clarified the assumptions underlying the model beginning on line 162 of the revised paper.

- 47. Fig. 3 b,d,f: I think it would be interesting to see/mark when the drug was added as well.**

We thank the reviewer for this comment. We wish to point out that, in these figure panels, we have aligned cell traces to the times at which we observed cytoplasmic condensation. As these times may differ between cells of the same population, and treatment occurred substantially (hours) before the leftmost points shown, we found that extending the figure leftward and adding markers to indicate when each cell was treated visually complicated the panel, as well as detracted from the features of interest shown. We certainly agree that clarifying this is helpful, and believe that the main point is that all cells have been continually treated by antibiotics for as long as they have been observed in these figures. We have therefore updated the relevant figure legend to better explain this, and hope that this sufficiently addresses the reviewer's comment.

- 48. The methods concerning microfluidics mention the flow rate in units of Pascal, which seems to be a device setting? Could the authors specify either a flow through in volume per time and channel dimensions and/or a shear rate?**

The reviewer is correct in thinking that the units of pressure are a device setting in the CellAsics platform. We have added pointers indicating that the 20 kPa pressure we set is equivalent to a flow rate of ~10 $\mu\text{L/hr}$ throughout the Methods section. We thank the reviewer again for their thoughtful comments, which have helped us to significantly improve the clarity of this work.

REVIEWER COMMENTS

Reviewer #1 (Remarks to the Author):

The revisions made by the authors satisfied all but one of my concerns raised during the first round review of the manuscript.

The one exception is regarding figure 6a where 1 x or 10 x MIC of ciprofloxacin and kanamycin, measured with cells treated with each drug alone, was used for all experiments, including those with co-treatment with glutathione, despite that co-treatment with glutathione increased MIC for both antimicrobials by >5-fold. I suggested that the authors use antimicrobial concentrations that are normalized to MIC to do the killing experiments. Otherwise, the comparison between samples with/without glutathione is much like giving a same, fix dose of drug to a 40-kg and a 200-kg patients and then conclude that the drug is more effective in the low-body weight patient without taking into account of body weight difference. The argument the authors listed in their rebuttal (response #9) is completely invalid. The MIC measurement applies about 10^5 cfu/ml, which gives a bacterial density that is not turbid at all. At 1 x MIC, both ciprofloxacin and kanamycin block bacterial growth (turbidity increase), they do little killing and definitely not killing all cells at such drug concentration. If addition of glutathione simply protects the inocula from being killed without allowing bacterial amplification, no turbidity increase and thus no MIC elevation would have been seen.

I suggest that the authors at least add data with normalized MIC to figure 6a and compare those without normalization to see whether the glutathione effect mainly derives from growth inhibition rather than from killing. Such a conclusion can be predicted from the authors own data in figure 6b.

I also suggest that the authors measure MIC for ciprofloxacin and kanamycin in the presence of DTT, mercaptoethanol, or α -tocopherol to see whether co-incubation with these compounds does not affect MIC. If so, it would provide an explanation for why these three compounds did not protect from killing (Figure 6g).

The data from glutathione and other antioxidants are not crucial for the main conclusion of this paper, you can leave them out if they do not fit or difficult to explain. Twisting them to fit will do more harm than good to your work.

Reviewer #2 (Remarks to the Author):

The authors have done an excellent job in addressing my concerns.

Reviewer #3 (Remarks to the Author):

The authors have done a very good job during this round of revision. I am very satisfied with the responses and the new experiences they have made.

Reviewer #4 (Remarks to the Author):

The authors have sufficiently addressed all my concerns and I recommend publication.

Matthias Koch

Response to reviewers

MS# NCOMMS-20-36348A

Wong, Stokes, Cervantes, Penkov, Friedrichs, Renner, and Collins, "Cytoplasmic condensation induced by membrane damage is associated with antibiotic lethality"

Summary of main changes.

We thank all the reviewers for their constructive and thoughtful comments on the paper. We have addressed the remaining points raised by Reviewer #1 through additional experiments and revisions. We would like to highlight the following key revisions:

(1) To address Reviewer #1's comment on distinguishing the potential effects of glutathione on kanamycin and ciprofloxacin-induced growth inhibition and killing, we have performed additional MIC and time-kill experiments. Further time-kill experiments, at larger kanamycin and ciprofloxacin concentrations for glutathione-pretreated cells, suggest that glutathione protection may arise from both decreasing growth inhibition and decreasing killing of individual cells.

(2) As suggested by Reviewer #1, we have performed additional MIC measurements for kanamycin and ciprofloxacin-treated cells in the presence of dithiothreitol, mercaptoethanol, and α -tocopherol. We found that the MICs were not significantly different from conditions in which no antioxidant was present, a finding which is consistent with the lack of protection observed in the corresponding time-kill experiments shown in Fig. 6g.

In the following, line numbers and citations refer to the revised version of the paper, and responses are indicated in blue font.

Reviewer #1.

1. The revisions made by the authors satisfied all but one of my concerns raised during the first round review of the manuscript.

We thank the reviewer for their important and constructive comments, which we hope to have adequately addressed in the revision.

2. The one exception is regarding figure 6a where 1 x or 10 x MIC of ciprofloxacin and kanamycin, measured with cells treated with each drug alone, was used for all experiments, including those with co-treatment with glutathione, despite that co-treatment with glutathione increased MIC for both antimicrobials by >5-fold. I suggested that the authors use antimicrobial concentrations that are normalized to MIC to do the killing experiments. Otherwise, the comparison between samples with/without glutathione is much like giving a same, fix dose of drug to a 40-kg and a 200-kg patients and then conclude that the drug is more effective in the low-body weight patient without taking into account of body weight difference. The argument the authors listed in their rebuttal (response #9) is completely invalid. The MIC measurement applies about 10E5 cfu/ml, which gives a bacterial density that is not turbid at all. At 1 x MIC, both ciprofloxacin and kanamycin block bacterial growth (turbidity increase), they do little killing and definitely not killing all cells at such drug concentration. If addition of glutathione simply protects the inocula from being killed without allowing bacterial amplification, no turbidity increase and thus no MIC elevation would have been seen.

We thank the reviewer for their clarification and agree that it would be interesting to separate the effects of glutathione on kanamycin- and ciprofloxacin-induced growth inhibition from those of killing. To address this, we have performed additional MIC and time-kill experiments for cells pretreated with 10 mM glutathione. Consistent with the data presented in Table R1 of the previous response letter, replicate MIC experiments showed that the MIC was increased ~10-fold by the presence of 10 mM glutathione for kanamycin, and ~3-

fold by the presence of 10 mM glutathione for ciprofloxacin; the empirical MIC ranges and working concentrations have now been added to Supplementary Table 1. In corresponding time-kill experiments, cells that were pretreated with 10 mM glutathione were treated with kanamycin or ciprofloxacin at concentrations relative to these increased MICs. We found that the presence of glutathione still led to increased survival relative to cases of no glutathione with kanamycin treatment in the range of 1× MIC and ciprofloxacin treatment at various treatment times in the range of 0.1× to 10× MIC; these data are now shown in Fig. 6a,b and discussed in further detail starting on line 291. Thus, these observations suggest that the observed glutathione protection may arise from both decreasing growth inhibition and decreasing killing of individual cells.

- 3. I suggest that the authors at least add data with normalized MIC to figure 6a and compare those without normalization to see whether the glutathione effect mainly derives from growth inhibition rather than from killing. Such a conclusion can be predicted from the authors own data in figure 6b.**

We thank the reviewer for this suggestion and, as detailed in response (1) above, have added data from assays with normalized MICs, for comparison to those without normalization. We agree that the conclusion might be predicted from the data in Fig. 6b and have added additional text and a reference to Fig. 6b beginning on line 293 to further emphasize this panel and its relation to the MIC concentrations used. We have also performed additional time-kill experiments to extend the right-hand side of the kanamycin plot in Fig. 6b, which has better revealed the range of kanamycin concentrations at which protection by 10 mM glutathione occurs. Finally, we have shown additional curves, corresponding to normalization with respect to increased MIC values in the presence of 10 mM glutathione, in Fig. 6b.

- 4. I also suggest that the authors measure MIC for ciprofloxacin and kanamycin in the presence of DTT, mercaptoethanol, or α -tocopherol to see whether co-incubation with these compounds does not affect MIC. If so, it would provide an explanation for why these three compounds did not protect from killing (Figure 6g).**

We have performed additional MIC experiments similar to response #1 above in order to address this point. We found, indeed, that the presence of 10 mM dithiothreitol, 10 mM mercaptoethanol, or 50 mM α -tocopherol did not change the range of empirically observed kanamycin and ciprofloxacin MICs, as now summarized in Supplementary Table 1 and beginning on line 330 of the revised paper. We believe that these results are consistent with the observation that these three compounds did not protect from the antibiotic treatments shown in Fig. 6g, as the reviewer correctly predicts.

- 5. The data from glutathione and other antioxidants are not crucial for the main conclusion of this paper, you can leave them out if they do not fit or difficult to explain. Twisting them to fit will do more harm than good to your work.**

We thank the reviewer for this thoughtful suggestion. We agree that the data from glutathione and other antioxidants are orthogonal to the main message of the work, which is that cytoplasmic condensation is a cell death phenotype in antibiotic-treated *E. coli* cells. Nevertheless, we do feel that the observation that glutathione pretreatment reduces the frequency of cytoplasmic condensation and lysis (Fig. 6c,h) could be interesting to readers, and our consideration of antioxidant perturbations is related to our observations of fluorescence in reactive metabolic byproduct-sensitive dyes (Fig. 4). We have therefore elected to retain these data in the main text, and we very much hope that our additional revisions better clarify our findings. We thank the reviewer again for their insightful comments on our work.

Reviewer #2.

- 6. The authors have done an excellent job in addressing my concerns.**

We thank the reviewer for their thoughtful comments, which have helped to strengthen our work significantly.

Reviewer #3.

- 7. The authors have done a very good job during this round of revision. I am very satisfied with the responses and the new experiences they have made.**

We thank the reviewer for their thoughtful comments, which have helped to strengthen our work significantly.

Reviewer #4.

- 8. The authors have sufficiently addressed all my concerns and I recommend publication.**

We thank the reviewer for their thoughtful comments, which have helped to strengthen our work significantly.

REVIEWERS' COMMENTS

Reviewer #1 (Remarks to the Author):

The authors have done a wonderful job in addressing my additional concerns. I now recommend publication.

Response to reviewers

MS# NCOMMS-20-36348B

Wong, Stokes, Cervantes, Penkov, Friedrichs, Renner, and Collins, "Cytoplasmic condensation induced by membrane damage is associated with antibiotic lethality"

Reviewer #1.

- 1. The authors have done a wonderful job in addressing my additional concerns. I now recommend publication.**

We thank the reviewer for their thoughtful comments, which have helped to strengthen our work significantly.